# Physiological Mechanisms by Which the Functional Ingredients in Beer Impact Human Health

**DOI:** 10.3390/molecules29133110

**Published:** 2024-06-29

**Authors:** Yawen Zeng, Hafiz Ghulam Muhu-Din Ahmed, Xia Li, Li’e Yang, Xiaoying Pu, Xiaomeng Yang, Tao Yang, Jiazhen Yang

**Affiliations:** 1Biotechnology and Germplasm Resources Institute, Yunnan Academy of Agricultural Sciences/Agricultural Biotechnology Key Laboratory of Yunnan Province, Kunming 650205, China; lixia_napus@163.com (X.L.); yangyanglie@163.com (L.Y.); puxiaoying@163.com (X.P.); yxm89ccf@126.com (X.Y.);; 2Department of Plant Breeding and Genetics, Faculty of Agriculture and Environment, The Islamia University of Bahawalpur, Bahawalpur 63100, Pakistan; ghulam.muhudin@iub.edu.pk; 3Key Laboratory of the Southwestern Crop Gene Resources and Germplasm Innovation, Ministry of Agriculture, Kunming 650205, China

**Keywords:** functional ingredients, actional mechanism, barley grains and grass, beer, malt, hops, functional foods, chronic diseases, human

## Abstract

Nutritional therapy, for example through beer, is the best solution to human chronic diseases. In this article, we demonstrate the physiological mechanisms of the functional ingredients in beer with health-promoting effects, based on the PubMed, Google, CNKI, and ISI Web of Science databases, published from 1997 to 2024. Beer, a complex of barley malt and hops, is rich in functional ingredients. The health effects of beer against 26 chronic diseases are highly similar to those of barley due to the physiological mechanisms of polyphenols (phenolic acids, flavonoids), melatonin, minerals, bitter acids, vitamins, and peptides. Functional beer with low purine and high active ingredients made from pure barley malt, as well as an additional functional food, represents an important development direction, specifically, ginger beer, ginseng beer, and coix-lily beer, as consumed by our ancestors ca. 9000 years ago. Low-purine beer can be produced via enzymatic and biological degradation and adsorption of purines, as well as dandelion addition. Therefore, this review paper not only reveals the physiological mechanisms of beer in overcoming chronic human diseases, but also provides a scientific basis for the development of functional beer with health-promoting effects.

## 1. Introduction

Refined staple foods damage the global economy and threaten human health. As a result, the global cost of five major chronic diseases’ treatment alone will reach USD 47 trillion from 2011 to 2030 [1]. The brewing ingredients of beer drunk 9000 years ago in early Holocene southern China may have included red rice, coix seeds, acorns, lily bulbs, and fungi [2]. Barley played an important role in health and civilization in the past, prompting human migration from Africa to Asia, and later to Eurasia. Barley and hops contain polyphenols with anti-bacterial, anti-inflammatory, anti-oxidative, anti-angiogenic, anti-melanogenic, anti-cancer, and anti-osteoporotic effects: these polyphenols are kaempferol, quercetin, tyrosol, ferulic acid, xanthohumol, isoxanthohumol, 8-prenylnaringenin, α-bitter acids like humulone, and β-bitter acids like lupulone [3]. Yeast transforms polyphenolic compounds in malt wort into healthy hydroxy-tyrosol and melatonin, as well as into probiotic drivers [4]. A general wish for healthier lifestyles has resulted in increased demand for functional beers with the health benefits and sensory adjustments of classical beer, which has broadened the market for the brewing industry [5]. The German Beer Purity Law from 1516 allows the use of only barley (*Hordeum vulgare* L.), hops (*Humulus lupulus* L.), yeasts, and water for beer brewing.

This ancient German Beer Purity Law has much scientific basis today. Beer is not only one of the oldest human inventions as a functional food but also the most common beverage worldwide, with 1.89 billion hectoliters consumed in 2022 at a value of USD 721.12 billion. The potential for functional beer expansion is endless when it comes to combining beer with herbs, spices, and other functional compounds [5]. While China has built a functional food industry of USD 1.4 billion on ginseng and notoginseng, we have forgotten that barley malt, one of the raw materials of brewed beer, has a value of USD 25 billion. Beer has the potential to be an excellent functional food in the future, based on its functional ingredient (barley malt) and traditional Chinese medicinal properties (from hops), especially its polyphenols and dietary fiber, with anti-inflammatory and anti-oxidant effects [6]. Beer phenolics derive from barley (lignans, alkylresorcinols) and hops (prenylflavonoids, stilbenes), with both contributing (phenolic acids, flavonoids); for instance, about 70% of beer polyphenols originate from barley malt, and the remaining 30% originate from hops. Nine components of phenolic acids have been identified in conventional beers, with ferulic acid as the most abundant phenolic acid, followed by caffeic, sinapic, p-coumaric, and vanillic acids [7,8]. Barley is rich in 30 ingredients that combat 28 chronic diseases, due to the molecular mechanism of barley grass (γ-aminobutyric acid (GABA), flavonoids, superoxide dismutase (SOD), potassium-calcium (K-Ca), vitamins, and tryptophan) and grains (β-glucans, polyphenols, arabinoxylan, phytosterols, tocols, and resistant starch) [1]. Purple barley, rich in phenols and flavonoids as well as anthocyanins, has a good anti-oxidant impact and α-glucosidase-inhibiting activity [9]. The six natural prenylated flavonoids (isoxanthohumol, isoxanthohumol-C, 8-prenylnaringenin, 6-prenylnaringenin, xanthohumol, and xanthohumol-C) of beer and hops vary from 0.04 to 3.2 μg/L [10].

Beer is a functional food against chronic diseases, which contains many functional ingredients, such as phenolic acids, polio phenol, quercetin, kaempferol, tyrosol, flavones, bitter acids (humulones and lupulones), vitamin B (pyridoxal (VB6), vitamin B12 (VB12), and folate), 8-prenylnaringenin, xanthohumol, minerals, and complex carbohydrates, except for ethanol [11]. The functional ingredients in beer can mitigate the negative effects of alcohol. Four compounds (total prenylated flavonoids, tyrosol, hydroxy-tyrosol, and alkyl-resorcinols) can act synergistically and trigger health effects [12]. Moderate beer consumption of up to 16 g alcohol/day for women and 28 g/day for men is associated with a decreased incidence of cardiovascular disease and overall mortality, among other metabolic health benefits [11,13], but the benefits may vary according to age, sex, genetics, and body type, as well as drug or supplement use [14]. There are about 17,000 factories producing craft beer that is unfiltered/unpasteurized around the world, and innovation concerns aspects such as ingredients, alcohol content, aging, and packaging. Craft beers are beverages rich in health compounds but with a reduced shelf life [15]. Non-alcoholic beer consumption is more effective than conventional beer in preserving the endothelial function and inhibiting thrombogenic activity, but conventional beer with high polyphenols induces greater increases in high-density lipoprotein (HDL) cholesterol levels [16].

When we consider obesity, diabetes, cardiometabolic diseases, and other chronic diseases, we forget that, in addition to glucagon-like peptide-1 (GLP-1) drugs (the Breakthrough of the Year), there are many functional foods, including barley and beer. It is necessary to clarify the co-evolutionary mechanism of disease resistance genes and human functional foods (beer, barley, and rice), as well as GLP-1 receptor agonists [17]. Although beer has contributed greatly to human health and global economic development, its negative effects have restricted the high-quality development of the industry. To date, there is a lack of systematic reports on the human health benefits of its active ingredients and the functional mechanism of barley and beer brewed from it, at home and abroad. The aim of this review is to provide new ways for pure barley beer and new functional beer to expand the development space for high-value-added functional foods, especially by offering new insights and theoretical support for the physiological mechanisms of beer’s active ingredients for benefitting human health. This review has important theoretical significance and high practical value for improving the proportion of barley malt in brewing and improving human health.

## 2. Functional Ingredients of Beer and Its Raw Materials

Beer quality is closely related to protein and its functional components. The quality traits of beer include flavor, texture, foam stability, gushing, peptides, and haze formation. There are 7113 proteins (4692 beer proteins and 3906 foam proteins), including lipid transfer protein (LTP), serpin, hordein, gliadin, and glutenin, which may help to evaluate the health risks of beer and promote healthy nutritional enrichment [18]. The foam-stabilizing proteins in beer from malted barley are protein Z4 and LTP, which display less foamability but greater foam stability [18]. The foam stability depends on the balance of foaming components (polypeptides, hop bitter acids, metal ions, melanoidins, ethanol, lipids, and detergents) [15].

### 2.1. Contribution of Hops and Barley to Beer Polyphenols

Hops are largely used in traditional medicine, with 1000 polyphenolic substances, especially proanthocyanidins (>55%), flavonoid glycosides (>28%), and polyphenols (40–140 mg/g); these became appreciated over the years for the bitterness and aroma they impart to beer [8]. Previously, the best sensory beer was found to be that with a higher content of lemon balm [5]. Phenolic anti-oxidants in beer may originate from lemon and hesperidin. These comprise 26 compounds belonging to the different phenolic classes of hydroxybenzoic, hydroxycinnamic, and caffeoylquinic acids, flavonoids, and prenylflavonoids [19].

Whether due to phenolic acids or flavonoids, hops and barley mostly produce the same functional composition in beer, though there are a few differences; for instance, stilbenes are mainly from hops. In Table 1, fourteen bioactive phenolic compounds’ contents are presented from hops and barley found in common beer. Both hops and barley can provide bioactive phenolic components, such as caffeic acid, chlorogenic acid, p-coumaric acid, ferulic acid, p-salicylic acid, syringic acid, gallic acid, protocatechuic acids, catechin, kaempferol, naringenin, naringin, quercetin, and rutin [1,8,20,21]. In Table 2, fourteen phenolic compounds’ contents are presented from hops found in common beer, for instance, gentisic acid, epigallocatechin, procyanidin B1, procyanidin B2, procyanidin C1, desmethylxanthohumol, isorhamnetin, isoxanthohumol, xanthohumol, total trans-stilbenes, 8-prenylnaringenin, trans-resveratrol, and trans-Piceid [8]. Furthermore, Table 2 shows eight phenolic compounds’ contents from barley found in common beer: sinapic acid, 2,4-dihydroxybenzoic, vanillic acid, flavonoids, myricetin, hesperidin, alkylresorcinols, and lignans [7,8].

### 2.2. Contribution of Functional Ingredients of Barley to Beer

The nutritional functional components of beer are mainly derived from barley sprouts. Table 3 shows the functional and nutrient compositions of beer and barley grains. Beer’s water content is from 88.5% to 97.7%. Barley grains are higher in both protein and nine mineral elements, as well as four functional components (polyphenols, flavonoids, ferric-reducing anti-oxidant power (FRAP), and 2,2-azino-bis (3-ethylbenzothiazoline-6-sulfonic acid) diammonium salt (ABTS)] than beer, and there are more than 13 nutritional functional differences between beer and barley grains. This table lists nine mineral elements of beer and barley grains. Among these, Se is very important for human health, and its deficiency can cause a serious immune deficiency, cognitive decline, and even mortality [5]. The highest proportion of anthocyanins (48.07 mg/100 g), total phenols (570.78 mg/100 g), and flavonoids (47.08 mg/100 g) are found in purple barley [9].

Beer is not only a popular functional beverage consumed in large amounts all over the world but also a source of carbohydrates, amino acids, minerals, vitamins, polyphenols (phenolic acids, flavonoids), and benzoic and cinnamic acid derivatives. Many research studies have focused on the identification of the functional ingredients in conventional beers. Table 4 lists the functional ingredients of beer in the order of phenolic acid (26) > flavonoids (13) > organic acid (12) > vitamins (10) > prenylflavonoids (5) = stilbenes (5) > amine (3) > dietary fiber (2) = phenolic alcohols (2) = melanoidins (2) > alkylresorcinols (1). Beer contains numerous categories of anti-oxidants, polyphenols, traces of group B vitamins, minerals (Se, Si, K), soluble fibers, melanoidins, and microorganisms [6]. In total, 13 components of flavonoids, 5 components of prenylflavonoids, 4 components of stilbenes, and 2 components of phenolic alcohols from hops have been identified in conventional beers [7,8,21]. Meanwhile, 17 components of phenolic acids from barley have been identified in conventional beers, with the most abundant being ferulic acid and procyanidin B. Moreover, 22 components of flavonoids and 12 components of lignan as well as alkylresorcinols from barley have been identified in common beers [1,8,21].

In addition, common beer is rich in vitamins and organic acids, including 14 vitamins and 13 organic acids [7,21]. Twenty-six samples of bottled beer contained significant amounts of biologically active iso-flavonoid phytoestrogens, with four iso-flavonoids ranging from 1.26 to 29 nmol/L (iso-flavonoid 0.19–14.99 nmol/L, daidzein 0.08–2.5 nmol/L, genistein 0.169–6.74 nmol/L, and biochanin A 0.820–4.84 nmol/L) [28].

### 2.3. Functional Beer

Whether in ancient times, now, or in the future, beer is seen as an important functional beverage. Functional beer is increasingly formulated for health benefits and a pleasant taste, such as by using ginger, olives, eggplant, lignans, green beer, and so on. The natural ginger beer market is dominated by the United States. Supplementing brewing wort with curcumin (25 μg/mL) can increase the beer’s total phenolic and flavonoid contents (shagaol, gingerone, zingerone), offering potential anti-biofilm and health benefits [31]. In a previous study, ginger beer produced with ginger bug and fermented for 14 days showed better volatile and phenolic compound profiles, physico-chemical parameters, microbial diversity, and sensory characteristics [32]. Meanwhile, olive leaves in beer (10 g/L) impart a sour/astringent taste and herbal aroma; however, 5 g/L may produce a pleasant sensory profile and can increase the oleuropein and 3-hydroxytyrosol of beers to support human health, especially when the boiling time favors the hydrolysis of oleuropein to 3-hydroxytyrosol [33]. Beer with added eggplant peel extract (delphinidin-3-rutinoside, delphininidin-3-glucoside, and delphinidin-3-rutinoside-5-glucoside) can increase anti-oxidant activity, based on its total phenolic content, from 0.426 to 0.631 mg/mL and total flavonoids from 0.065 to 0.171 mg/mL [34]. Lignans are plant phenols for human health, and the addition of spruce knot chips or extracts during wort boiling produces beer with lignan contents ranging from 34 to 174 mg/L [35]. Additionally, the formulation of green beers can increase the phenolic and anti-oxidant activity, e.g., turmeric/black pepper/aroma hops = 1.5:1.5:1 or 5:2:3 [36].

Functional beer may be improved through novel brewing approaches, which have become a new field of beer brewing and human health. Recently, characterizations of the functional ingredients and anti-oxidant properties of beers have noted added fruits, vegetables, herbs, and natural foods. Beers with lemon juice, raspberry syrup, orange juice, and grapes expand the range of consumers to women and those who dislike bitterness [5]. Moreover, fruit beers obtained through the addition of 1 of 18 fresh fruits during the fermentation process result in significant enrichment in phenolics and total flavonoids, as well as stilbene molecules’ compounds in beers, and similar results have been found for 20 beers produced through the addition of vegetables, herbs, and natural foods [8].

Beer is a worldwide functional beverage, based on the addition (herbs, fruits, fungi, probiotics, etc.) or removal (gluten or carbohydrates) of certain compounds for human health [5]. Gluten-free beer is responsible for the market expansion to gluten-intolerant people, and low-carbohydrate beers have been better-selling beverages than regular beers [5]. In addition to foreign countries that develop functional beer with functional food materials, China also uses 34 functional food materials (astragalus, barley malt, barbary wolfberry, rugosa rose, Rhizoma gastrodiae, Polygonatum sibiricum, hawthorn, jujube, lily, fructus mori, yam, papain, Lonicerae japonicae, Polygonatum odoratum, Herba lophatheri, semen cassia, almond, oyster, sea buckthorn, Siraitiae fructus, mulberry leaf, chrysanthemum, Perilla frutescens, Pueraria lobata, honey, citrus peel, Mentha, raspberry, Dendrobium candidum, ginseng, Ganoderma lucidum, cornel, honey, etc.) to develop new functional beers.

### 2.4. Special Beer

Rice beer is traditionally brewed and consumed by Southeast Asian countries. The metabolite profiles of rice beer consist of 18 saccharides, 18 organic acids, 11 sugar alcohols, 8 amino acids, and 1 vitamin and nutraceutical compound, along with thiocoumarine, carotene, oxazolidine-2-one, and acetyl tyrosine. The alcohol content of rice beer varies from 9.41 to 19.33%, phenolics vary from 2.07 to 5.40 mg gallic acid/mL, 1,1-Diphenyl-2-picrylhydrazyl radical 2,2-Diphenyl-1-(2,4,6-trinitrophenyl)hydrazy (DPPH·) varies from 1.94–4.14, and ABTS+ varies from 1.69–3.91 mg of Vc/mL [37]. In addition, there are special beers, such as rye, oat, millet, potato, quinoa, tartary buckwheat, jasmine, green tea, coffee, cocoa bean, kelp, pepper, mungbean, Moringa stenopetala, etc. It has been found that consuming non-alcoholic beer before exercise can help maintain electrolyte homeostasis during exercise [38]. In a previous study, industrial Radler beer with green tea microbeads was found to be the best sensory beer with the least bitterness and a strong pleasant herbal taste [5].

## 3. The Health Effects of Raw Materials

### 3.1. Barley Grains with 28 Health Effects

Barley grains play an important role in health and civilization, and prompted human migration from Africa to Asia, and later to Eurasia [1]. Barley represents the highest comprehensive utilization of a grain crop, largely for forage feed, wine raw materials, healthy food, functional food, ornamental weaving, and Chinese medicine. To date, 28 human health effects have been reported for barley grains. In addition to 22 chronic diseases for which it can be applied for prevention and treatment, as reported by Zeng et al. [1], barley has also been found to have further effects in recent years, such as reducing the symptoms of COVID-19, treating fatty liver and Alzheimer’s disease, improving glucolipid modulation, having cardiometabolic effects, and supporting the efficacy of prebiotic formulations.

#### 3.1.1. Barley Grains Have 22 Health Effects

Zeng et al. [1] summarized the health effects of barley for 22 chronic diseases, for which it may be applied for prevention and treatment, including anti-diabetes effects, anti-obesity, anti-cancer, anti-oxidants, improved gastrointestinal function [39], anti-inflammation, lowering the blood pressure, hepatoprotection, immunomodulation, alleviating atopic dermatitis, preventing cardiovascular diseases, treating hyperlipidemia, antiaging, cardio-protection, anti-fatigue, optimizing cholesterol, improving bowel health, treating metabolic syndrome, preventing heart disease, reducing chronic kidney disease [40], accelerating wound healing, and decreasing stroke and cholelithiasis activities.

Above all others, barley and adlay are the most common functional foods, with the use of 110 kinds of homologous substances, in China, both barley grasses and grains, to prevent and control more than 22 chronic diseases in humans and animals, based on the molecular mechanisms of barley grass (GABA, flavonoids, SOD, K-Ca, vitamins, and tryptophan) and grains (β-glucans, polyphenols, arabinoxylan, phytosterols, tocols, and resistant starch). Their uses provide a reference for the development of human functional foods and animal husbandry production of functional feed [1].

Undoubtedly, the results of research to date support highland barley, barley, or its malt’s selection for the 2023 and 2024 Editions of the *Adult Dietary Guidelines for Hyperlipidemia, Hypertension, Diabetes, Obesity, Chronic Kidney Disease, Hyperuricemia and Gout*, as well as the *Dietary Guidelines for Growth Retardation and Obesity in Children and Adolescents* released by the National Health Commission of the People’s Republic of China.

#### 3.1.2. Controlling COVID-19

The role of barley malt was reported in COVID-19 prescribing in eight provinces in China, and the authors noted that barley grass powder has a certain effect on COVID-19 treatment. Moreover, it was reported that barley-based remedies could significantly enhance the blood oxygen saturation and reduce fatigue in COVID-19 patients [41]. The ricin-based peptide from barley was able to inhibit Mpro in vitro with an IC50 of 0.52 nM, and it had low or no cytotoxicity at up to 50 µM, which suggested its therapeutic potential against SARS-CoV-2 [42]. Barley and its functional foods used enhancing human immunity and control COVID-19, especially the quercetin in barley, are inhibitors of the main protease of SARS-CoV-2. Such new perspectives facilitated caramel barley malt’s inclusion in the Diagnosis and Treatment Plan for COVID-19 (trial version 10) in 2023 of the National Health Commission of the People’s Republic of China. Furthermore, it has been shown that Persian barley water can reduce the length of hospital stays, fever, the erythrocyte sedimentation rate, C-reactive protein, and creatinine among hospitalized COVID-19 patients with moderate severity [43,44], supporting the treatment protocol approved by Iran’s Ministry of Health and Medical Education. The Iranian regimen is as follows: Ficus carica, Vitis vinifera, Safflower, Cicer arietinum, Descurainiasophia seeds, Ziziphus jujuba, chicken soup, barley soup, rose water, saffron, and cinnamon spices. This appears to be effective in the treatment of symptoms as well as inflammatory biomarkers, such as C-reactive protein, in COVID-19 patients [45].

#### 3.1.3. Treating Fatty Liver

Highland barley as a functional food reduces the incidence of fatty liver. The extract of highland barley, Monascus purpureus, improves non-alcoholic fatty liver disease (NAFLD) through various pathways and targets of the body’s metabolism, and it can be used as a functional food for the treatment of liver disease and lipid metabolism disorders [46]. Highland barley β-glucan alleviated Western diet-induced NAFLD through preventing fat accumulation by increasing energy expenditure, with a mechanism associated with changes in hepatic bile acid composition [47].

#### 3.1.4. Treating Alzheimer’s Disease

Highland barley contains wide bioactive nutrients, such as carbohydrates, polyphenols, minerals, vitamins, phenolic, flavonoids, and β-glucan, which contribute to many health benefits, such as treating Alzheimer’s disease, anti-bacterial, anti-inflammatory, anti-cancer, anti-diabetic, anti-obesity, anti-fatigue, anti-aging, and preventing hyperglycemia, hyperlipidemia, and heart disease, as well as cancers [48].

#### 3.1.5. Glucolipid Modulation

A large intake of a high-energy diet leads to obesity, and highland-barley-based functional foods may help manage hyperlipidemia and have anti-obesity effects, which work through human brain control of excessive appetite. In this way, barley functional foods and genes help overcome obesity. Moreover, barley is rich in a variety of functional ingredients with health-promoting effects, especially those that improve glucolipid modulation mechanisms [49]. In one study, barley (80%) reduced blood concentrations of total cholesterol, low-density lipoprotein, and triglycerides in fattening pigs; however, the HDL content increased extremely significantly to 1.04 mmol/dm^3^ [50].

#### 3.1.6. Cardiometabolic Effect

In previous research, bread containing whole-grain red sorghum and barley flours enhanced the plasma total polyphenols and anti-oxidant status, and its consumption modulated biomarkers of cardiometabolic health [51].

#### 3.1.7. Prebiotic Formulation

Barley by-products are important raw materials for the production of melanoidins. Barley melanoidins prevent lipid peroxidation and oxidative damage of DNA, and they induce anti-oxidant activity and anti-microbial, anti-hypertensive, anti-allergenic, and prebiotic properties [52]. High phenolic compounds and β-glucans and tocols in barley are among the important substrates of functional foods, such as probiotic formulations [53].

#### 3.1.8. Neuroprotection

Barley grains are rich in protein, fiber, minerals, and phenolic compounds, which have potency against neurodegenerative diseases through anti-oxidant, anti-inflammatory, and vasodilatating activities, along with restoring neurochemical alterations [54].

### 3.2. Hops for Human Health

Hops have a characteristic aroma and bitter taste, based on functional ingredients such as humulones, lupulones, 8-prenylnaringenin, 6-prenylanaringenin, xantho-humol, 6,8-diprenylnaringenin, and 8-geranylnaringenin, and their health activities are based on antibiotic, anti-bacterial, anti-molds anti-viral, anti-fungal, neuro-preventative, estrogenic, and anti-inflammatory properties. These are especially effective in preventing chronic diseases, such as cancer, arthritis, insulin sensitivity, type II diabetes, metabolic syndrome, and menopause [8,55]. The estrogenic effects of hop prenylflavonoids place them among the most potent phytoestrogens identified so far. In a previous study, the yield of 8-prenylnaringenin was 29 mg/100 g of the product, which may have alleviated the symptoms of menopausal disorders [56] and had anti-cancer or estrogenic activity [57]. Xantho-humol in hops has synergistic effects against aging, diabetes, inflammation, microbial infection, proliferation, and cancer [58]. As a result, hops contribute greatly to human health, and as the raw material for brewing they also increase the health effect of beer.

## 4. Structure–Function Claim Substantiation for Beer

### 4.1. Health Effects of Beer

The 26 health effects of beer are the result of the combined effect of its raw materials brewed into active ingredients (Table 5; Figure 1). Numerous studies have shown that the functional ingredients in beer have many health benefits. Above all, polyphenols have more than 8000 different types identified so far, which include phenolic acids, flavonoids, stilbenes, and lignans. Table 3 lists the functional ingredients of beer, which include 26 phenolic acids, 13 flavonoids, 5 prenylflavonoids, and 5 stilbenes. In addition to the many health effects of phenolic acids and flavonoids, prenylflavonoids from hops can prevent osteoporosis, inhibit cytochrome-P450-mediated activation of procarcinogens, have antiproliferative effects on cancer cell lines, and have anti-angiogenic activity [8]. The stilbene from hops has anti-platelet, anti-inflammatory, estrogenic, cardioprotective, anti-tumor, and anti-viral properties. Lignans from barley have anti-tumor, anti-oxidative, anti-viral, anti-bacterial, anti-fungal, estrogenic, cardioprotective, and cardiovascular activities [8].

The metabolites of hydroxy-tyrosol and tyrosol exhibit strong anti-cancer, neuroprotective, cardioprotective, anti-atherogenic, anti-oxidant, and endocrine effects [8]. Moreover, the melatonin contained in beer can provide health benefits, due to its neuroprotective, anti-cancer, anti-oxidant, anti-inflammatory, and immunomodulatory properties, especially in cardiovascular diseases, osteoporosis, control of hypertension, and constipation [59,60]. Four phytochemicals from barley (tocopherols, tocotrienols, carotenoids, phytosterols) exhibit strong anti-oxidant, anti-proliferative, and cholesterol-lowering activities, which can prevent some human chronic diseases, such as cancer, cardiovascular disease, diabetes, and obesity [8].

#### 4.1.1. Cardiovascular Disease Prevention

Cardiovascular disease has emerged as the leading cause of death worldwide. There are many functional components of beer that prevent cardiovascular disease. Undoubtedly, beer has protective effects on the cardiovascular system and prevents cancer through the alcohol and polyphenols consumed with a moderate beer intake [61,62]. Beer phenolics in free and bound forms mainly originate from barley malt and roasted malts [63]. These reduce human cardiovascular disease by improving vascular elasticity, mediating vascular dilation, and significantly increasing apolipoprotein A1 levels [64]. Moreover, the high folic acid level in beer reduces cardiovascular disease among alcohol consumers [65].

#### 4.1.2. Anti-Cancer

Global cancer deaths total nearly 10 million; however, there are many functional components of beer that can prevent cancer. Phenols and melanin in beer have anti-oxidant properties, which can reduce the content of free radicals in the human metabolism and have clear effects against cancer and cardiovascular diseases [66]. A moderate intake of beer provides polyphenols, vitamins, and fiber, a low intake (3 mg/kg) inhibits the formation of lesions, polyps, and tumors, and a high intake (300 mg/kg) protects against early cancer [67]. The prenylated flavonoids of beer and hops have therapeutic potential for cancer and metabolic syndrome [10].

#### 4.1.3. Anti-Diabetes

Unhealthy diets drive the exponential growth of diabetes patients worldwide, but some functional components of beer can prevent diabetes. A high-energy diet and lack of human brain control of excessive appetite lead to diabetes [1]. Isomaltulose and resistant maltodextrin in alcohol-free beer improve insulin resistance in patients with type 2 diabetes and overweight or obesity [68]. A daily intake of 330 mL alcoholic beer can change the lipid profile and insulin sensitivity of adult men [69]. Beer with low sugar and alcohol contents offers a functional food for diabetics [70].

#### 4.1.4. Lipid Deposition Prevention

Human obesity outbreaks, owing to the psychological pursuit of pleasurable tastes, arise from lack of a healthy diet. In this context, a moderate intake of beer prevents lipid deposition in vessel walls, and does not increase body weight in obese healthy individuals [71]. The soluble nitrogen content (74.98%) of barley protein is a dietary secretion-stimulating supplement for obesity management [72]. Moderate consumption of beer improves the lipid profile of postmenopausal women, although its potential for preventing cardiometabolic effects warrants further investigation [73].

#### 4.1.5. Antioxidation

Anti-oxidation is at the core of delaying aging. The anti-oxidants of polyphenols and melanoidins in beer have the effect of protecting against oxidative damage [74]. The majority of beer’s anti-oxidant activity (55–88%) originates from ferulic acid (50%) and the five phenolics (syringic acid, catechin, caffeic acid, protocatechuic acid, and epicatechin) [75]. Non-alcoholic beer is more effective in preventing oxidative stress than conventional beer [16]. A moderate intake of beer increases the anti-oxidative properties of HDL and facilitates cholesterol efflux [71].

#### 4.1.6. Anti-Inflammation

An anti-inflammatory diet is popular nowadays. Angiogenesis and inflammation signaling are targets of beer polyphenols on vascular cells [76]. Bitter iso-α-acids in beer from hops suppress both lipid accumulation and brain inflammation [77]. Hop cones are rich in phenolic compounds and have anti-cancer, anti-oxidant, and anti-inflammatory activities, especially the effects of anti-platelet drugs [78]. The hop metabolite 8-prenylnaringenin stimulates angiogenesis, whereas xanthohumol and isoxanthohumol have anti-angiogenic and anti-inflammatory effects [76].

#### 4.1.7. Immunomodulation

Immunomodulation can help prevent all diseases worldwide. Beer degradation products have anti-inflammatory, anti-coagulant, and anti-oxidant actions and regulate glucolipid metabolism, as well as leading to anti-cancer outcomes, reduced cardiovascular events, and regulation of metabolic syndrome [79]. Beer contains melatonin, which is a molecule with anti-oxidant, onco-static, immunomodulatory, and cytoprotective properties [60]. The essential oils, bitter acids, and flavonoids of hop cones for brewed beer have anti-oxidant and immunomodulatory as well as neuroprotective effects [80].

#### 4.1.8. Improves Gastro-Intestinal Health

One-fifth of the world’s population suffers from gastroenteritis, and bacteria are the main culprit. Beer polyphenols can reach the large intestine and interact with the colonic microbiota [81]. Gastric emptying in full-intensity, low-carbohydrate, and low-alcohol beer determines the plasma ethanol response in healthy young people, i.e., the most significant negative correlation between plasma ethanol at 15 min and gastric emptying after low-alcohol (AUC 0–180 min for blood glucose) was greater for low-alcohol than low-carbohydrate beer [82]. Moderate beer drinking is conducive to intestinal health due to the higher concentration of butyric acid in beer consumers [83].

#### 4.1.9. Cardio-Protection

One-third of global deaths are caused by heart disease. There may be minor benefits associated with drinking beer in terms of reducing coronary disease [84]. The polyphenols and melanoidins in beer have cardioprotective and anti-cancer effects, regulate the intestinal microbiota and metabolites, and have anti-bacterial and anti-inflammatory properties [74].

#### 4.1.10. Longevity

Anti-oxidation is the key to greater longevity. The relationship between health and longevity associated with beer drinking was reported in 1884. The main nutrients of beer are polyphenols (30% from hops, 70% from malt), amino acids, carbohydrates, minerals, and vitamins, and there are anti-aging, anti-oxidant, and proliferative effects of brewing components on D-dSCs and Caco-2 cells [85]. Treatment with ethanol and beer before irradiation can enhance longevity in mosquitoes [86]. The iso-α-acids of beer prevent dementia and anti-aging by suppressing neuroinflammation and improving cognitive function [87]. VB2 has anti-oxidant, anti-aging, anti-inflammatory, anti-nociceptive, and anti-cancer properties [88].

#### 4.1.11. Improves Skin Health

More than 200 million people around the world experience repeated extreme pruritus. The health effects of beer substances on skin include the treatment of atopic eczema, contact dermatitis, hyper-pigmentation diseases, skin infections, skin aging, skin cancer, and photo-protective measures [4]. Malt and beer by-products (caramalt, aromatic malt, roasted malt, malt sprouts, dark beer spent grain) are anti-oxidants for whitening cosmetics and so on [89]. Ferulic acid from by-products of beer is an anti-oxidant with cosmeceutical properties. Nano-ferulic acid promotes the regeneration of the skin [90].

#### 4.1.12. Prevents Alzheimer’s Disease

Alzheimer’s disease brings a heavy economic burden to countries around the world. Regular beer consumption can prevent Alzheimer’s and neurodegenerative diseases by reducing the aluminum, and the mineral homeostasis imbalance in the brain [91]. In a previous study, a hop extract had γ-secretase inhibitory activity and significantly reduced Aβ production in cultured cells and Alzheimer’s disease in mice [92].

#### 4.1.13. Improves Metabolic Syndrome

The global prevalence of metabolic syndrome varies widely, especially in adults, but is approaching 50%. Polyphenols and bitter acids (iso-α-acids) in hop cones are very effective in treating metabolic syndrome, especially through hypo-glycemic, anti-hyperlipidemic, anti-obesity, lipid metabolic, and glucose tolerance activities [93]. The health effects of bitter acids and xanthohumol in beer on chronic diseases are specifically focused on inflammatory and immune diseases, obesity, metabolic disorders, and cancer prevention [94].

#### 4.1.14. Prevents Osteoporosis

About 2300 million people worldwide suffer from osteoporosis. Polyphenols of non-alcoholic beer can improve bone health and body water in postmenopausal women [95]. Beer is more effective than spirits in increasing bone mineral density, such as the effects of Si in beer, which reveals that its components, except ethanol, may contribute to bone health [96]. The functional components of beer are polyphenols, folic acid, and phytoestrogens, which can reduce osteoporosis [97].

#### 4.1.15. Improves Cognition

In recent years, the prevention and control of cognitive disorders have gradually become a major focus of public health. Dementia and cognitive decline are global public health problems. Daily intake of iso-α-acids in beer from hops suppresses inflammation in the hippocampus, improves visual learning, and reduces cognitive decline, and when paired with a high-fat diet, it enhances the production of inflammatory cytokines and chemokines as well as improving memory as a function of dopamine release in mice, meaning that it could be effective for improving the working memory in human dementia patients [77]. The supplementation of bitter acids at 35 mg/day can stimulate the vagus nerve and enhance cognitive function [98].

#### 4.1.16. Anti-Depressant

About 1 billion people around the world suffer from mental disorders. Hop β-bitter acid, especially hoperone, has potential anti-depressant effects in vitro [99]. Iso-α-acid intake through beer prevents brain aging and neurological disorders, which are related to the effects of the gray matter volume and cognitive function for the Brain Healthcare Quotient [100].

#### 4.1.17. Improves Fatigue or Mood

Fatigue and the mood are associated with sleep. Drinking non-alcoholic beer in daily life can maintain a good mood, including reducing anxiety, depression, and fatigue, while improving vigor, sleep quality, and absolute presenteeism [101]. Hop bitter acids in beer might be beneficial for cognition and the mood state [98]. Alcoholic beer consumption showed a J-shaped relationship with self-perceived, physical, mental, and social–emotional health, with better values at moderate levels [102].

#### 4.1.18. Blood Pressure Regulation

There are 1.3 billion people living with hypertension worldwide. Beer is major food source of isoxanthohumol, a precursor of 8-prenylnaringenin. The phytoestrogens can reduce perimenopausal symptoms, especially through moderate non-alcoholic beer consumption, which improves the lipid profile and decreases the blood pressure in postmenopausal women [103]. Five beers led to a significant decrease in systolic blood pressure, heart rate, and self-reported feeling of palpitations [104].

#### 4.1.19. Prevents Neurodegenerative Disease

Neurological disorders have become a highly serious health problem. Beer extracts prevent neurodegenerative diseases by regulating adenosine receptor expression and protecting glioma and neuroblastoma cells from oxidation, which supports the benefits of beer consumption for health [105]. Low/moderate consumption (1–2 drinks per day of beer) reduces the risk of cardiovascular and neurodegenerative disease [106]. Moderate beer consumption was associated with a protective cardiovascular function and a reduction in the development of neurodegenerative disease [10], and had a neuroprotective effect through a possible depletion of Aβ aggregation in the brain [107].

#### 4.1.20. Hepatoprotection

There are 1.3 billion people with liver disease worldwide. A moderate beer intake can have hepatoprotective effects due to the flavonoids in hops. In previous research, between eight beers, it was found that the imperial red ale exhibited the highest anti-oxidant properties and increased the enzymatic and non-enzymatic redox status after CCl_4_ insult [108]. The mechanisms for the anti-atherosclerotic effect of xanthohumol in beer have been proven to involve decreasing pro-inflammatory factors and improving hepatic lipid metabolism via adenosine 5′-monophosphate-activated protein kinase (AMPK) activation [109].

#### 4.1.21. Promotes Sleep

Sleep is the cornerstone of good health. Sleep deprivation affects the physiological functions of humans. Melatonin is a hormone secreted in the pineal gland with several functions, especially the regulation of the circadian sleep cycle and biological processes [59]. The consumption of non-alcoholic beer at dinnertime helps to improve the quality of sleep at night [110]. In previous research, the sleep-promoting effects of α-acids, β-acids, and xanthohumol in beer from hops, especially Simcoe water extract and Saphir ethanol extract, modulated GABAergic signaling to improve sleep-related behaviors, including sleep duration [111].

#### 4.1.22. Heart Failure or Stroke Prevention

Stroke has become the second-leading cause of death in the global population. Oxidative stress of polyphenols plays a central role in cardiovascular disease (myocardial infarction, angina pectoris, cardiac ischemia, heart failure, chronic arrhythmias, and stroke), especially by modulating the transcriptomic response of heart oxidative stress through myocardial ischemia in hypercholesterolemia; meanwhile, a dose-dependent up-regulation of electron transport chain members and the down-regulation of spliceosome-associated genes have been linked to beer consumption [112]. Low beer consumption acts against cardiovascular disease through its impact on low-grade chronic inflammation [113].

#### 4.1.23. Prevents Gallstone Disease

More than 100 million people worldwide have gallstones, and alcohol consumption of beer can decrease the risk of gallstone disease. In previous research, there was a dose-dependent linear risk reduction and a weakened linear trend between alcohol consumption levels less than and greater than 28 g/day [114]. Regular moderate beer drinking lowers the bile concentration and affects cholesterol levels, which lowers the likelihood of gallstones [115].

#### 4.1.24. Reduces Kidney Stones

Kidney stones, a urinary tract disease, are among the oldest and most prevalent medical conditions around the world. Water can mix with blood to produce secretions and to perform metabolic reactions. Barley water is an excellent solution for kidney stones and cysts, in both children and adult age groups; it should be consumed daily till the urine infection subsides [116]. In line with this, beer is a diuretic that may aid in the passage of tiny stones that are less than 5 mm in size. Furthermore, high intakes of Ca, K, and fluids have been shown to be associated with a lowered risk of kidney stones, and one bottle of beer consumed per day can reduce the risk by 40% [117]. Supplementation of beer with pine shoots can facilitate the dissolution of kidney stones, based on its phenolic acids and flavonoids, which have anti-bacterial, expectorant, and analgesic properties; this beer may be used as an antiseptic for respiratory tract, urinary tract, and kidney infections [118].

#### 4.1.25. Wound Healing Acceleration

Dandelion addition improved anti-oxidant beer properties via chlorogenic, caffeic, ferulic, and chicoric acid [119]. Oxidative stress inhibits wound healing, but phytochemicals, especially chicoric acid, have anti-oxidant and scavenge reactive oxygen species, thereby promoting wound healing [120].

#### 4.1.26. Prebiotic Action

Prebiotics are functional foods for the growth and expansion of healthy bacteria in the large intestine and metabolism to produce short-chain fatty acids. Lactobacillus fermentum TIU19 isolated from Haria beer can be explored as a potential probiotic with antagonistic activity against multi-drug resistance uro-pathogenic *E. coli* and *E. faecalis* [121]. The polyphenol degradation products from beer have prebiotic action and may combat intestinal dysbiosis [6].

**Table 5 molecules-29-03110-t005:** Functional ingredients of beers and barley for prevention of chronic disease.

Preventive Chronic Disease	Functional Ingredients in Grains/Grass [1]	Functional Ingredients in Beers	Reference in Beers
Cardiovascular disease prevention	β-glucans, arabinoxylan, lignans, tocols, polyphenols, phytosterols, folate/K, Ca, saponarin, SOD, VB1, vitamins (A, C, E), GABA, tryptophan	polyphenols, melanin, folate, Mg, prenylated flavonoids, melanoidins	[11,61,62,63,65,66,74]
Anti-cancer	β-glucan, phenolics, arabinoxylan, lignan, phytosterols, resistant starch/Alkaline, flavonoids, chlorophyll, tricin, SOD	Prenyl-flavonoids, Se, melanin, polyphenols, tyrosol, vitamins, bitter acids, peptides, stilbene, hydroxy-tyrosol	[8,10,67,94,122,123]
Anti-diabetes	β-glucan, phenolic compounds, polysaccharide, phytosterols, tocols, resistant starch/Ca, Saponarin, dietary fiber, AMPK, SOD, polyamines, GABA	Isomaltulose, phenolic acids, Se, resistant maltodextrin, bitter acids, xanthohumol, VB8, peptides	[8,68,69,93,122,123,124]
Lipid deposition prevention or anti-obesity	β-glucan, polysaccharide, starch, polyphenols, dietary fiber, α-tocopherol, tocols, phytosterols (saponarin, 2″-O-glycosyl iso-vitexin)	soluble nitrogen, Mg, iso-α-acids, xanthohumol, polyphenols	[11,72,125]
Antioxidation	polyphenols, GABA, VE, phenolics, anthocyanin, polysaccharide, SOD, tocotrienol/lutonarin, γ-tocopherol, saponarin, orientin, chlorophyll, iso-orientin, glutathione, flavonoid	polyphenols, Se, VB2, iso-α-acids ferulic acid, melanoidins, flavonoids	[8,74,75,80,88]
Anti-inflammation or allergic rhinitis alleviation	β-glucans, vanillic acid, lignans, arabinoxylan/Chlorophyll, saponarin, SOD, tryptophan, GABA	iso-α-acids, peptides, VB2, 8-prenylnaringenin, isoxanthohumol, polyphenols, melanoidins, xanthohumol	[59,60,76,77,78,88,123]
Immunomodulation	β-glucans, arabinoxylan/polysaccharide, GABA	melatonin, bitter acids, flavonoids, bioactive peptides	[60,80,122]
Improve gastrointestinal	β-glucans/dietary fiber, Se, GABA	polyphenols, butyric acid, flavonoids; melanoidins	[59,60,81,83]
Cardio-protection	β-d-glucan/K, GABA	polyphenols, peptides, VB6, melanoidins, flavonoids	[8,74,126]
Longevity or Anti-aging	Excavate functional components	polyphenols, minerals, vitamins, Iso-α-acids, flavonoids	[85,87]
Improve skin health or atopic dermatitis	GABA, extract P/metallothionein, SOD	flavonoids, by-products, melanoidins	[4,89]
Prevent Alzheimer’s disease	flavonoids, minerals, vitamins, phenolic acids in barley grains [48]	Si, hops extract, phenolic acids	[8,91,92]
Improve metabolic syndrome	β-glucans in barley grains	polyphenols, bitter acids, xanthohumol	[94]
Prevent osteoporosis or bone injury recovery	Excavate functional components. Ca	Si, polyphenols, VB9, phytoestrogens, melanoidins	[59,60,95,96,97]
Improve cognition	GABA, K, and SOD in barley grass	Iso-α-acids, Se	[77,123,127]
Antidepressant	GABA, saponarin, vitamins, and minerals in barley grass	β-bitter acid, iso-α-acids, phenolic acids	[8,99,100]
Improve fatigue or mood	Excavate components/lutonarin, saponarin	bitter acids	[98,101,102]
Blood pressure regulation	β-glucans/saponarin; lutonarin, K, Ca, GABA	8-prenylnaringenin, melatonin, peptides, VB6	[103,122,126]
Prevent neurodegenerative disease	protein, fiber, minerals, phenolic compounds in barley grains	flavonoids, minerals, melanoidins, bitter acids, peptides	[54,59,60,100,106,107,122,128]
Hepatoprotection	β-glucans, phenolics, pentosan/saponarin, SOD, GABA [47]	flavonoids, xanthohumol, Se	[108,109,123]
Promote sleep	GABA, Ca, K, VC, and tryptophan in barley grass	melatonin, α-acids, β-acids, xanthohumol	[59,110]
Heart failure or stroke prevention	β-d-glucan, phenolics, tocols, linoleic acid, extract P, low protein, folate in barley grains	polyphenols	[112,113]
Prevent gallstone	Excavate functional components	alcohol, ascorbic acid, Ca	[114]
Reduce kidney stones	β-glucans in barley grains	Ca, K, Se, fluid selenium,	[117,123]
Wound healing acceleration	β-glucans in barley grains	caffeic acid, ferulic acid, chicoric acid	[119,120]
Prebiotic action	melanoidins, tocols, phenolic compounds, β-glucans in barley grains [52,53]	polyphenols, fiber, melanoidins	[6]

### 4.2. Beer Ethanol for Human Health

Non-alcoholic beer promotes antioxidation [16], improves bone health [95], maintains mood states [98], regulates blood pressure [103], promotes sleep [110], etc. Decreased risk (17~21%) of dementia was associated with maintaining mild (<15 g per day) to moderate (15–29.9 g per day) ethanol consumption [129]; however, ethanol exceeding 105 g per week (i.e., <15 g per day) was associated with a reduced risk of death related to heart attack, stroke, angina, or cardiovascular disease. Therefore, it is recommended to drink beer with less than 30 g of ethanol per day, 5% ethanol in beer equivalent to 600 g of beer.

Table 3 shows that the sugar content in beer is 2.1–5.3%; beer with low sugar (≤0.75 g/100 mL) and alcohol (<1.2%) content is a functional food for diabetics [70]. Moderate intake of beer prevents lipid deposition in blood vessels and does not lead to obesity [71]; however, problems with obesity caused by overconsumption of beer may be related to other high-energy foods and gastrointestinal effects [83].

### 4.3. Beer’s Threats to Human Health

#### 4.3.1. Hyperuricemia

Beer has 26 health effects against 26 chronic diseases in the human body (Table 5), which are highly similar to the effects of barley (Figure 1); however, the interaction effects between excess adiposity and alcohol use are positively associated with hyperuricemia [130]. The d-amino acids in beer are the oxidation of Fe^2+^ to produce hydroxyl radicals, leading to DNA damage and the formation of a large number of purine bases [129]. Uric acid in humans is the end product of purine metabolism. Due to the loss of hepatic uricase activity during evolution, it is associated with oxidative stress and inflammation, as well as nephrolithiasis [131]. The purine in beer can be catabolized into uric acid, leading to hyperuricemia and gout [132]. Among the 63 associated foods previously studied, beer and liquor with the strongest urate-raising effect were associated with a 1.38 μmol/L increase in serum urate per serving per week, equating to a 9.66 μmol/L (0.16 mg/dL) increase per daily serving [133].

#### 4.3.2. Low-Purine Beer

Beer with low purine expands the range of consumers to people with hyperuricemia and gout. The purine content in beer can be reduced by enzymatic and biological degradation or adsorption methods [132]. The purine content in beer after treatment with adenine deaminase and guanine deaminase enzymes was significantly reduced; the two enzymes can work at 5.0–8.0 pH and retain >50% activity at 40 °C, which offers an effective route to low-purine beer production [134]. Dandelion addition improved the beer’s anti-oxidant properties and inhibited uric acid production thanks to xanthine oxidase from chlorogenic, caffeic, ferulic, and chicoric acid [119].

#### 4.3.3. Hyperuricemia and Gout Diet Beer

According to the 2024 Editions of the released by the National Health Commission of the People’s Republic of China, there is a high number of raw materials for treating hyperuricemia and gout, which may be consumed via beer. Chicory (7) > orange peel (4) = coix seed (4), and four functional foods (Poria cocos, pueraria, honeysuckle, and lily) are mentioned twice, and a further 17 functional foods (Amomi fructus, ginger, chrysanthemum, hawthorn, Chinese yam, jujube, ginseng, Astragalus, etc.) are mentioned once. These materials’ addition has the potential to develop functional beer for preventing gout and hyperuricemia.

## 5. Physiological Mechanisms of Beer Affecting Human Health

### 5.1. Polyphenol Mechanism

Polyphenols in beer have 24 health effects, such as cardiovascular disease prevention, anti-cancer, anti-diabetes, lipid deposition prevention, anti-obesity, anti-oxidation, anti-inflammation, immunomodulation, improving gastrointestinal health, cardio-protection, anti-aging, improving skin health, atopic dermatitis alleviation, improving metabolic syndrome, preventing osteoporosis, bone injury recovery, blood pressure regulation, neuroprotection, hepatoprotection, promoting sleep, heart failure or stroke prevention, wound healing acceleration, prebiotic action, etc. (Table 5, Figure 2). Therefore, the best functional beer has the highest content of total phenols and greatest anti-oxidant activity [5].

Polyphenols can act in the stimulation of β-oxidation and adipocyte differentiation inhibition as well as counteract oxidative stress by modulating physiological and molecular pathways of energy metabolism [135]. Specifically, phenolic compounds in beer are important for human health since they have anti-tumor and anti-oxidant activities [136]. The inhibition of intact human vascular smooth muscle cells through ecto-alkaline phosphatase activity by polyphenolic compounds and polyphenol-containing beers may contribute to their cardiovascular protective effects [137]. Xanthohumol and 8-prenylnaringenin from beer-derived polyphenols can ameliorate diabetic-associated metabolic disturbances by regulating glucose and lipid pathways, which resulted in skeletal muscle AMPK signaling pathway activation, suppressed lipogenesis, prevented body weight gain, and improved the plasma lipid profile, with significant improvement in insulin resistance and glucose tolerance [138].

Most importantly, a moderate beer consumption supplies polyphenols and phenolic acids with anti-inflammatory and anti-oxidant activities, as well as improves gastrointestinal functions in human health, increases gut microbiota diversity and fecal alkaline phosphatase activity, significantly increases butyric acid, and reduces the ammonium content [139,140]. The polyphenol intake gained through moderate beer consumption has health effects including reducing osteoporosis and cardiovascular risk and relieving vasomotor symptoms, especially 8-prenylnaringenin, 6-prenylnaringenin, and isoxanthohumol, with intracellular estrogen receptors leading to the modulation of gene expression and increasing sex hormone plasma concentrations [141]. In previous research, a 15% colored malt was sweet, had a deep color, had high bitterness and turbidity, and had an alcohol content of 6.2–6.8%, along with the highest polyphenols (453.8 mg/L) and anti-oxidant activity (840.1 µmol/L) and the lowest foam stability [142].

#### 5.1.1. Phenolic Acid Mechanism

Phenolic acids play a role in anti-cancer, anti-diabetes, anti-inflammatory, and anti-oxidative effects, improve skin health, Alzheimer’s and Parkinson’s disease, and depression, ameliorate ischemia and reperfusion injury, etc. [8] (Table 5, and Figure 2). Phenolic acids supply the anti-oxidant and anti-inflammation properties of the H-atom through a mechanism of free radical scavenging [143].

Specifically, gallic acid has many health effects in bacterial or viral infections, cancer, inflammation, neuropsychological disorders, gastrointestinal conditions, and metabolic disease. It inhibits bacterial growth by altering the membrane structure, and inhibits cancer cell growth, oncogenes, and matrix metalloproteinases’ expression by targeting different signaling pathways in apoptosis, increasing reactive oxygen species’ production [144]. Protocatechuic acid has anti-allodynic and anti-hyper-analgesic effects through adenosine triphosphate-sensitive potassium (KATP) channel activation related to A1 receptor stimulation [145]. Gentisic acid regulates the miR-19b-3p/RAF1 axis to mediate the extracellular-signal-regulated kinase (ERK) pathway and inhibit the development of rheumatoid arthritis [146]. Chlorogenic acids regulate key targets in the tumor necrosis factor (TNF) signaling pathway, inhibit the polarization of microglia to the M1 phenotype, and ameliorate neuroinflammation-induced cognitive dysfunction in mice [147]. Vanillic acid has anti-bacterial and anti-biofilm activity against carbapenem-resistant E. hormaechei, which work by rupturing the cell membrane integrity, decreasing intracellular ATP, the pH, and the membrane potential, inhibiting biofilm formation, and killing cells within biofilms [148]. Caffeic acid phenylethyl ester has several pharmacological effects, such as anti-bacterial, anti-tumor, anti-oxidant, and anti-inflammatory activities; lowering the temperature, adding biosurfactants and sodium deoxycholate, and the presence of Cu^2+^ increases the binding force between caffeic acid phenylethyl ester and hemoglobin [149]. Ferulic acid has anti-inflammatory, analgesic, anti-radiation, anti-cancer, and immune-enhancing effects; it can cause mitochondrial apoptosis through intracellular reactive oxygen, inducing autophagy, a series of cell targets, and the regulation of tumor cell signaling pathways [150]. Syringic acid has anti-oxidant and anti-inflammatory properties that reduce the progression of Parkinson’s disease, along with neuroprotective effects [151]. 4-Hydroxyphenylacetic acid is the major intestinal metabolite of kaempferol and polymeric proanthocyanidins; its anti-bacterial effects can cause cell death through cell membrane damage and decrease the expression of three virulence factors [152]. Sinapic acid is a potent anti-oxidant used for the treatment of cancer, infections, oxidative stress, and inflammation; its anti-cancer mechanism works through the regulation of multiple proteins (CTNNB1, PRKCA, CASP8, SIRT1) and cytochrome enzymes (CYP1A1, CYP3A4) [153].

#### 5.1.2. Flavonoid Mechanism

Refined staple foods damage the global economy and threaten human health. More than half of the global population will be overweight or obese by 2035, and over 1.3 billion people will have diabetes by 2050. Flavonoids in beer have 22 health effects, such as cardiovascular disease prevention, anti-cancer, anti-diabetes, lipid deposition prevention, anti-obesity, anti-oxidation, immunomodulation, anti-inflammation, improving gastrointestinal health, cardio-protection, anti-aging, improving skin health, improving metabolic syndrome, blood pressure regulation, neuroprotection, hepatoprotection, promoting sleep, anti-bacterial, anti-viral, and so on (Table 5 and Figure 2).

In particular, the prenylated flavonoids in beer from hops, such as 8-prenylnaringenin and 6-prenylnaringenin, through spontaneous cyclisation into isoxanthohumol, and subsequently demethylation by gut bacteria, in combinations of metabolism involving hydroxylation, sulfation, and glucuronidation, result in an unknown number of isomers [154]. Xanthohumol in hops treats a variety of cancers, due to the inhibition of cancer cell growth and proliferation by regulating multiple signaling pathways (Akt, AMPK, ERK, insulin like growth factor binding protein 2 (IGFBP2), NF-κB, STAT 3) and proteins (Notch1, caspases, MMPs, Bcl-2, cyclin D1, oxidative stress markers, tumor-suppressor proteins, and miRNAs) [58]. Catechins enhance skeletal muscle performance, due to epicatechin gallate, with the strongest differentiation, significantly reduce adhesion force and stiffness and enhance C2C12 cell differentiation [155]. The anti-bacterial activity works through hydrogen bonding and hydrophobic and electrostatic interaction between epicatechin gallate and anionic carboxymethyl Poria cocos polysaccharide [156].

Most importantly, rutin, along with neuroprotection, has an anti-oxidative impact through up-regulating the expression of P-ERK and Nrf2 proteins in the ERK/Nrf2 pathway [157]. Rutin and quercetin with anti-oxidants can reduce the formation of protein oxidation products and produce the highest clearance rates for DPPH (62.74%) and ABTS^+^ (71.14%) [158]. Quercetin has anti-inflammatory, anti-oxidative, and osteo-protective properties, which reduce 13 indicators (arthritis scores, paw swelling, histopathological scores, interleukin-1β, interleukin-6, interleukin-17, tumor necrosis factor-α, monocyte chemotactic protein-1, C-reactive protein, malondialdehyde, reactive oxygen species, thio-barbituric acid reactive substances, nuclear factor kappa B) and increase 6 indicators (interleukin-10, catalase, glutathione peroxidase, SOD, glutathione, and heme oxygenase-1) [159]. Kaempferol alleviating sepsis can reduce inflammatory reactive oxygen species (ROS) production and cell apoptosis by acting on the HIF-1, NF-κB, and PI3K-Akt signaling pathways [160].

Moreover, genistein can induce bone remodeling, which involves osteoblasts, osteoclasts, and osteocytes and different modes of intracellular signaling, through Wnt/β-catenin pathway activation [161]. Luteolin can hinder ovarian cancer cell proliferation and activate the PI3K/AKT pathway, leading to apoptosis [162]. Apigenin has anti-oxidative, anti-inflammatory, and anti-cancer activities by inhibiting GLUT-1 mRNA and protein expression in head and neck cancers [163]. Myricetin inhibited the proliferation and induced the apoptosis of H1975 cells by regulating the expression of MMP 1, MMP 3, MMP 9, and PIK3R1 genes and various pathways [164]. Naringin induced osteoblastic differentiation of BMSCs by activating the BMP2/Runx2/Osterix signaling pathway and promoted the regulation of estrogen receptor pathway protein expression, a significant in vivo ectopic osteogenic effect [165].

### 5.2. Melatonin Mechanism

Melatonin in beer has 14 health effects, such as cardiovascular disease prevention, anti-cancer, antioxidation, anti-inflammation, immunomodulation, atopic dermatitis alleviation, improved gastrointestinal health, cardio-protection, preventing osteoporosis, bone injury recovery, blood pressure regulation, neuroprotection, promoting sleep, prebiotic action, etc., (Table 5 and Figure 2). The melatonin mechanism involves free radical scavenging properties and complex intracellular signaling pathways, limiting the entry of tumor cells into the vascular stream and their distribution in other organs and systems, which blocks the growth of metastases in locations far from the original tumor [59,60]. ROS-dependent formation of 2-hydroxymelatonin from melatonin was the major pathway in cancer cells, which could serve as an index for the endogenous reactive oxygen level and oxygen-carrying capacity of hemoglobin in human blood [166].

Melatonin with anti-oxidant and anti-inflammatory properties alleviated hepatic damage aggravated by PCB126-induced ROS-dependent NET formation by suppressing excessive ROS production [167]. Melatonin may act indirectly on the immune system through the circadian clock to regulate food allergies [168]. Melatonin protects against ketorolac-induced gastric mucosal toxic injuries through a molecular mechanism associated with the modulation of arylakylamine N-acetyltransferase activity. In one study, a correlation between a depleted gastric melatonin level and ulcer formation unveiled a novel ulcerogenic mechanism [169]. The sleep rhythms of GW117 with anti-depressant action may be due to melatonin system-mediated activation of the Wnt/β-catenin signaling pathway [170].

### 5.3. Bitter Acid Mechanism

Bitter acids in beer have 15 health effects, such as anti-cancer, anti-diabetes, lipid deposition prevention, anti-obesity, anti-oxidation, anti-inflammation, allergic rhinitis alleviation, immunomodulation, anti-aging, improved metabolic syndrome, improved cognition, anti-depressant, improved fatigue or mood, neuroprotection, promoting sleep, etc. (Table 5 and Figure 2).

Specifically, bitter taste receptor activation by matured hop bitter acids drives a downstream Ca response and cholecystokinin production in enteroendocrine cells, as well as activating the gut–brain axis [171]. Consuming non-alcoholic beer in daily life can maintain a good mood state, since hop bitter acids improve the mood and reduce peripheral symptoms as well as enhancing stress-resilience-related hippocampal dopaminergic activity in humans [101]. Iso-α-acids in beer suppress hippocampal microglial inflammation and improve cognitive decline; moreover, matured hop bitter acids activate the vagus nerve and suppress neuronal damage and depression-like behavior induced by inflammation [172]. The sedation and anti-depressant mechanisms of hop bitter acid involve the activation of neuron-like Ca^2+^ channels by lupulones and tricyclolupones [99].

Bitter acids in beer enhance memory and cognitive functions through norepinephrine neurotransmission and vagus nerve stimulation. Iso-α-acids enhance hippocampus-dependent memory and prefrontal cortex-associated cognitive function through dopamine neurotransmission activation [98]. The iso-α-acids from the hop-derived bitter acids of beer improve spatial and object recognition memory, and they may contribute toward improving obesity-induced cognitive impairments [125]. The hop beta-acids mimic the action mechanism and the spectrum of ionophores and are a useful functional product of ruminants that inhibit rumen bacteria in the classical Gram-positive cell envelope [173].

### 5.4. Mineral Mechanism

Minerals facilitate various electrical and chemical processes. Minerals in beer have 15 health effects, such as cardiovascular disease prevention, anti-cancer, anti-diabetes, lipid deposition prevention, anti-obesity, anti-oxidation, longevity, anti-aging, Alzheimer’s prevention, preventing osteoporosis, bone injury recovery, improving cognition, neuroprotection, hepatoprotection, preventing gallstones, reducing kidney stones, etc. (Table 5 and Figure 2).

Se is an indispensable trace element (50–200 μg/day) for human health. More than 40 diseases are highly related to Se deficiency; for instance, Se supply can maintain glucose and lipid homeostasis in type 2 diabetes patients, and its deficiency or excess leads to β-cell dysfunction [123]. Se-rich soy peptides can prevent liver damage caused by heat stress and exercise fatigue by increasing the glutathione content and glutathione-peroxidase activity in rat livers, and protect rat livers by regulating the NF-κB/IκB pathway and preventing the release of interleukin-1β, interleukin-6, and tumor necrosis factor α. Anti-cancer SeNPs enhanced the autophagic ability of cancer cells by activating the ROS-mediated JNK pathway and inhibiting the PI3K/Akt/mTOR pathway [123]. Fermented beer enriched with organic Se (0.378 mg/kg) may be produced among functional foods [174]. Ca may alter the composition of bile by preventing the reabsorption of secondary bile acids in the colon, thus reducing the deoxycholate and cholesterol contents of the bile [129].

Beer contains Si and hop compounds that can play a major role in preventing brain disorders and acts as a potential instrument for protecting against neurodegenerative disease progression [91]. Al (NO_3_)_3_ induces metal imbalance, inflammation, and anti-oxidant damage in mouse brains by significantly blocking the effects of silicic acid and beer, especially when connecting the pro-oxidant markers with the brain Al content, while the brain Zn and Cu levels are closer to anti-oxidant enzyme expression [175]. The addition of 150 g persimmon fruit per 10 L of water could enrich the nutritional (polyphenol, Mg, K, and Ca), organoleptic, and anti-oxidant potentials of beer [176]. Mg^2+^ addition resulted in the pH of the fermenting wort decreasing quickly, an increase in the level of L-lactic acid, and increased concentrations of volatile compounds; meanwhile, Zn supplementation resulted in a decrease in the L-lactic acid content and a higher pH in the beer [177]. Light-to-moderate consumption of beer components (calcium ionophore) prevents coronary endothelial dysfunction associated with hyperlipemia-induced cardiovascular risk factors by counteracting vascular oxidative damage and preserving the Akt/eNOS pathway [178]. The use of Se-biofortified barley grain as a raw material can produce Se-enriched beer [179].

### 5.5. Vitamin Mechanism

Vitamins help cellular enzymes to regulate metabolic reactions. Vitamins in beer have 11 health effects, such as cardiovascular disease prevention, anti-cancer, anti-diabetes, anti-oxidation, anti-inflammation, cardio-protection, longevity, anti-aging, preventing osteoporosis, bone injury recovery, blood pressure regulation, etc. (Table 5 and Figure 2). Beer is rich in vitamins, such as the vitamin B family (thiamin (VB1), riboflavin (VB2), niacin (VB3), nicotinamide, pantothenic acid (VB5), VB6, biotin (VB7), inositol (VB8), folate, VB12, ascorbic acid (Vc), and fat-soluble vitamins (A, D, E, K)) (see Table 4).

Specifically, ginsenoside Rb1-induced restoration of redox homeostasis was mediated by targeting riboflavin transporters and riboflavin kinase [180]. Nicotinamide may reduce the release of pro-inflammatory mediators by inhibiting the MAPK and AKT/NF-κB signaling pathways and may ultimately alleviate lung injury [181]. Pyridoxal-5-phosphate in cardio-protection and vasorelaxation may be a result of increased expression of KATP channels and H_2_S production [126]. Myo-inosito may act as an insulin sensitizer through insulin-resistant tissues, such as PCOS-endometrium and SMIT-1, provoking AMPK activation and elevated GLUT-4 levels, thereby increasing glucose uptake by human endometrial cells [124]. Folate plays a protective role against atherosclerosis by regulating DNA methylation, ARID5B expression, and monocyte subsets [182]. High-dose ascorbic acid has a cytotoxic effect on myelodysplastic syndrome tumor cells, inhibiting cell proliferation and increasing apoptosis [183]. The protective effect of gallstones for vitamin C modulates the hepatic and biliary pathways of cholesterol homeostasis by promoting the conversion of cholesterol to bile acids through liver 7α-hydroxylation [129].

Moreover, after one month of drinking 830 mL of alcoholic beer every day, nine indicators (uric acid, anti-oxidative capacity, SOD, glutathione reductase, total cholesterol, HDL cholesterol, apolipoprotein-AI, LDL cholesterol, and apolipoprotein B) of 160 male volunteers increased, while vitamin B12 and fibrinogen decreased; this indicates that beer consumption is significantly correlated with uric acid and anti-oxidative capacity changes [184]. The consumption of industrial beer to prevent cardiovascular disease reduced the serum homocysteine (6.50~7.35 µmol/L) and increased folic acid (3.46~3.94 ng/mL); meanwhile, craft beer increased gamma-glutamyl transpeptidase (16.6~18.6 U/L) and reduced VB6 (20.9~16.9 ng/mL) [185].

### 5.6. Active Peptide Mechanism

Beer proteins can represent an excellent source of bioactive peptides. Peptides are used as a raw material for human muscles, blood, hormones, and new tissues. Exogenous bioactive peptides have properties, such as anti-cancer, anti-hypertensive, anti-diabetic, anti-inflammatory, immunomodulatory, anti-microbial, neuroprotective, cardiovascular protective, and opioid [122]. Peptides in beer have eight health effects: anti-cancer, anti-diabetes, anti-inflammation, allergic rhinitis alleviation, immunomodulation, cardio-protection, blood pressure regulation, and neuroprotection (see Table 5).

Specifically, the inhibitory peptides of the enzyme dipeptidyl-peptidase IV are a pharmacological target in type 2 diabetes therapy since it is involved in the degradation of the insulinotropic incretin hormones [122]. The bioactive peptides have anti-inflammation and gut health properties thanks to reducing TNF-α, NF-κB, and TLR4, improving IgA production and the intestinal morphology, and increasing the villi surface area and goblet cell diameter [186]. Beer quality is based on the flavor, texture, foam stability, gushing, and haze formation from beer peptides and 7113 proteins [17,20]. Over 1900 protein groups in the beer proteome were identified when applying IPG–IEF and LC–MS/MS [187]. Antibodies in PhIP-Seq data with beer peptides serve as excellent records of environmental exposures and immune responses, based on high abundance, relative stability, and easy accessibility in peripheral blood [188]. Proteins, peptides, and amino acids were assumed to form disulfide bonds with polyfunctional thiols in malt and hops, especially with the release of thiols through the reduction of disulfide-bonded thiols during fermentation [189]. The anxiolytic and antidepressant protein hydrolysates and peptides can exert effects through neurotransmitter systems, neurotrophic functions, neurons, nerves, and HPA axis mechanisms [190]. Bioactive peptides are short amino acid sequences against a variety of human diseases, including anti-cancer, anti-hypertensive, anti-diabetic, anti-inflammatory, anti-microbial, immunomodulatory, neuroprotective, and cardiovascular protective activities [122]. Fifty peptides in Tsingtao draft beer were identified, especially LNFDPNR and LPQQQAQFK peptides, which could bind angiotensin-converting enzyme and dipeptidyl peptidase IV tightly through hydrogen bonding and hydrophobic interaction, as well as raising their inhibitory activity [191].

In short, barley grains can prevent and control more than 20 chronic diseases in humans or animals, due to the molecular mechanisms of β-glucans, polyphenols, arabinoxylan, phytosterols, tocols, and resistant starch [1]. Meanwhile, barley grass powder can prevent and control more than 20 chronic diseases in humans and animals, thanks to the molecular mechanisms of GABA, flavonoids, SOD, K-Ca, vitamins, and tryptophan [192]. Beer contributes to cell protection and health-promoting effects based on many functional components against chronic diseases, such as polyphenols (xanthohumol), phenolic acids, melatonin, minerals, bitter acids, kaempferol, quercetin, tyrosol, flavones, xanthohumol, 8-prenylnaringenin, the vitamin B complex, citric acid, Vc, silicic acid, etc. [60,193]. The prevention and treatment of 26 human chronic diseases by consuming beer are the result of the comprehensive effect of barley grains and its grass, as well as hops, due to the molecular mechanisms of polyphenols (phenolic acids, flavonoids), melatonin, minerals, bitter acids, vitamins, and peptides (Table 5 and Figure 2). Beer styles show wide variation in color, flavor, and clarity, though the major flavor compounds are isomerized alpha acids and phenolic compounds [192]. First, the polyphenol mechanism (phenolic acids, flavonoids) of beer in preventing and treating 22 human chronic diseases involves the interaction between polyphenols of barley grains and flavonoids of barley malt and hops; flavonoids (especially saponarin and lutonarin) in barley grass have 11 similar health effects [194]. Second, the melatonin mechanism of beer in preventing and treating 14 human chronic diseases involves the interaction between the melatonin and GABA, as well as tryptophan, of barley malt; in the body, melatonin is synthesized from tryptophan, leading to competitive O-demethylation and C6-aromatic hydroxylation pathways [195]. Tryptophan in barley grass has four similar health effects [192]. Third, 15 health effects of the mineral mechanism of beer arise from the combined actions of barley malt, hops, and water. The K–Ca mechanism in barley grass has seven similar health effects [192]. The 11 health effects of the vitamin mechanism of beer originate from the combined actions of barley malt and hops, and the vitamin mechanism in barley grass has 7 similar health effects [192]. Fourth, there are 15 health effects of the bitter acid mechanism of beer from hops, and the GABA mechanism in barley grass has 13 similar health effects [192].

These results provide a reference for the development of human functional food and animal husbandry production functional feed [1], and demonstrate that nutritional therapy is the best solution to human chronic diseases. High nutrition increases the crop yield required and leads to human obesity; these effects create adversity, which should increase the crop functional composition and human intelligence. The transition from high-yield agriculture and green agriculture to functional agriculture will present a breakthrough across domains, and much related scientific research needs to be carried out in the future.

## 6. Conclusions and Future Perspectives

Refined staple foods damage the global economy and threaten human health. Nutritional therapy, such as through beer, offers a solution to chronic human diseases. Although common beers and special beers are changing rapidly, functional beer can be limited to four ingredients: barley, hops, water, and yeast. Beer represents the functional ingredients of barley malt and hops fermented into more absorbable small-molecule active ingredients under the action of yeast, and may constitute an excellent functional food in the future.

First, beer, as a complex of two functional ingredients, barley malt and hops, is rich in active ingredients. More than 1000 polyphenolic substances in hops contribute to the bitterness and aroma of beer, and about 70% of beer polyphenols originate from barley malt, with the remaining 30% from hops. Hops and barley can provide 8 phenolic acids and 6 flavonoids; gentisic acid and 10 flavonoids comes from hops, and 3 flavonoids come from barley malt. Barley grains are higher in protein, nine mineral elements, and four functional components than beer, and there are more than 13 nutritional functional differences between beer and barley grains.

Second, the health effects of beer against 26 chronic diseases are highly similar to those of barley. These include cardiovascular disease prevention, anti-cancer, anti-diabetes, lipid deposition prevention, anti-obesity, anti-oxidation, anti-inflammation, immunomodulation, improved gastrointestinal health, cardio-protection, longevity, improved skin health, preventing Alzheimer’s disease, improved metabolic syndrome, preventing osteoporosis, improved cognition, anti-depressant, improved fatigue or mood, blood pressure regulation, preventing neurodegenerative disease, hepatoprotection, promoting sleep, heart failure or stroke prevention, preventing gallstone disease, reduced kidney stones, wound healing acceleration, prebiotic action, etc.

Third, beer has similar effects and molecular mechanisms to prevent and treat human chronic diseases to those of barley. Barley combats 28 chronic diseases thanks to the molecular mechanisms of barley grass (GABA, flavonoids, SOD, K-Ca, vitamins, and tryptophan) and grains (β-glucans, polyphenols, arabinoxylan, phytosterols, tocols, and resistant starch). Beer prevents 26 chronic diseases, due to the molecular mechanisms of polyphenols (phenolic acids, flavonoids), melatonin, minerals, bitter acids, vitamin, and peptides. The main sources of these functional ingredients are as follows: four materials for making beer provide mineral elements, three materials (excluding water) provide polyphenols (phenolic acids, flavonoids), vitamins are obtained from barley malt and hops, active peptides and melatonin are sourced from barley malt, and bitter acids are derived from hops. While barley and its malt were selected for the eight Dietary Guidelines released by the National Health Commission of the People’s Republic of China, beer was not included. Low-purine beer and gout-preventive beer stand as important development directions for functional beer, such as chicory beer, orange beer, ginger beer, hawthorn beer, jujube beer, ginseng beer, and coix-lily beer, like that developed by our ancestors ca. 9000 years ago.

In sum, the complex and variable interactions of beer and its materials (barley, hops, water, and yeast, potentially with functional foods’ addition) require a comprehensive discussion to correctly address beer’s role in the management of human chronic diseases. The similarities and differences between the molecular mechanisms of barley, hops, and beer provide fruitful grounds for future research and development, for the prevention and treatment of chronic diseases. Beer can harm the human body, with the excessive consumption of beer with high purine content leading to hyperuricemia and even gout, since it has the strongest urate-raising effect of 63 purine foods. Low-purine beer can be produced through enzymatic and biological degradation and adsorption of purines, as well as dandelion addition. Therefore, functional beer with low purine and high active ingredients made from pure beer and barley malt, in terms of functional foods’ addition, offers an important development direction, to improve the added value and enhance the market competitiveness of beer. In order to reduce the cost, brewers should produce beer by using rice, wheat, and corn as the raw materials, with low active ingredients. These beers will seriously reduce the negative health effects of beer drinking, and lower prices and high taxes are recommended.

## Figures and Tables

**Figure 1 molecules-29-03110-f001:**
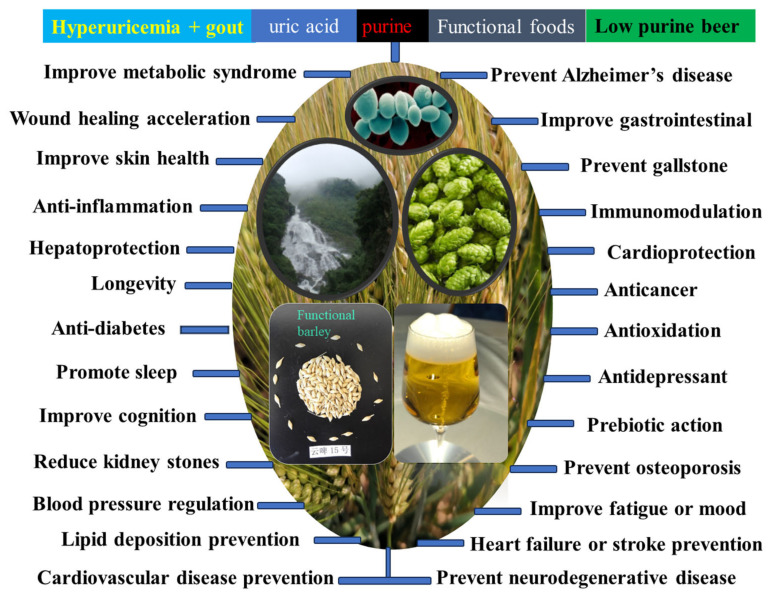
The health effects of beer against 26 chronic diseases are highly similar to those of barley. These 26 health effects of beer are the result of the combined effects of barley malt, hops, yeast, and water, brewed into active ingredients. In particular, the more than 20 health benefits of barley grains and their grass powder coincide with the effects of beer. However, the high purine content of beer can easily cause hyperuricemia and gout. The development of low-purine pure barley beer and functional beer is an important direction for functional foods. These research results support the theory that barley malt, hops, and beer brewing as a main nutritional therapy is the best way to treat human chronic diseases, and that there is only one cell disease theory for functional foods in humans.

**Figure 2 molecules-29-03110-f002:**
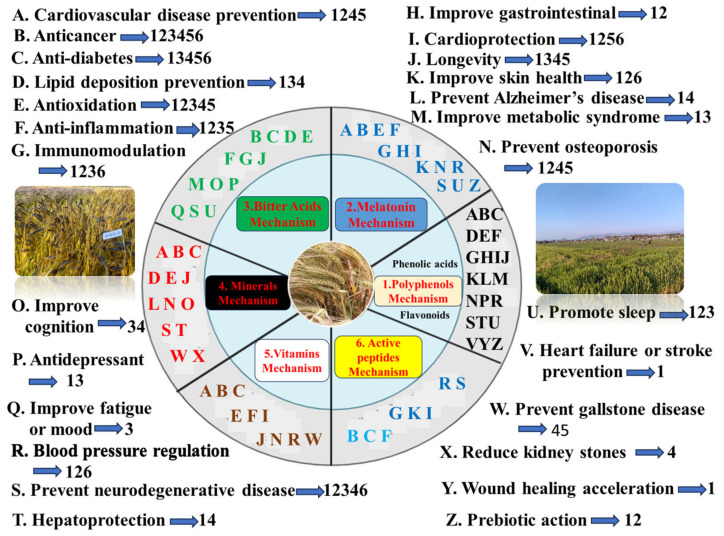
Action mechanisms of six functional ingredients in beer for human health. There are some differences in the molecular mechanisms of the six functional components of beer in affecting human health, and the health effects are as follows: polyphenol mechanism, 22 health effects (phenolic acids, 11 effects; flavonoids, 22 effects) > bitter acid mechanism, 15 health effects ≥ mineral mechanism, 15 health effects > melatonin mechanism, 14 health effects > vitamin mechanism, 11 health effects > active peptide mechanism, 8 health effects.

**Table 1 molecules-29-03110-t001:** Fourteen phenolic compounds in common beer from hops and barley.

Compound (Molecular Formula) Phytochemical Name	Hops (mg/100 g)	Barley (mg/100 g)	Reference
Caffeic acid (C_9_H_8_O_4_) 3,4-Dihydroxycinnamic acid	0.01–15.8	0.17 ± 0.01 *	[1,8]
Chlorogenic acid (C_16_H_18_O_9_) 3-O-Caffeoylquinic acid	0.47–163.7	0–9.84	[8,21]
p-Coumaric acid (C_9_H_8_O_3_) 4-Hydroxycinnamic acid	0.01–28.8	0.17–58.3	[8]
Ferulic acid (C_10_H_10_O_4_) 4-Hydroxy-3-methoxycinnamic acid	1–10	0.59–4.25	[8]
p-Salicylic acid (C_7_H_6_O_3_) 4-Hydroxybenzoic acid	187.0 ± 1.0	0.58–2.67	[8]
Syringic acid (C_9_H_10_O_5_) 4-Hydroxy-3,5-dimethoxybenzoic acid	3–1290	0.1–91.6	[8,20]
Gallic acid (C_7_H_6_O_5_) 3,4,5-Trihydroxybenzoic acid	8–341	0.1–136.6	[8]
Protocatechuic acid (C_7_H_6_O_4_) 3,4-Dihydroxybenzoic acid	42–225	0.14 ± 0.05 *	[7,8]
Catechin (C_15_H_14_O_6_) (2S,3R)-2-(3,4-dihydroxyphenyl)chroman-3,5,7-triol	1.2–56.1	0.1–10.5	[1,7,8]
Kaempferol (C_15_H_10_O_6_) 3,5,7-Trihydroxy-2-(4-hydroxyphenyl)-4H-chromen-4-one	0.44–49.4	1.27–19.2	[1,8]
Naringenin (C_15_H_12_O_5_) 5,7,4′-Trihydroxyflavanone	3.9–11.0	4.7–50.2	[8]
Naringin(C_27_H_32_O_14_) hydroxy-2-(4-hydroxyphenyl)-2,3-dihydrochromen-4-one	1.7–3.9	0.77–6.97	[8]
Quercetin (C_15_H_10_O_7_) 3′,4′,5,7-Tetrahydroxyflavan-3-ol	1.03–111.8	2.0–8.7	[8,20]
Rutin(C_27_H_30_O_16_) Quercetin 3-O-rutinoside	61–88	1.4–11.8	[8]

Note: * For fresh weight; the rest is dry weight.

**Table 2 molecules-29-03110-t002:** Phenolic compounds in beer from hops or barley.

Hops Compound (mg/100 g)			Barley Compound (mg/100 g)		
(Molecular Formula)	[Reference]	Contents	(Molecular Formula)	[Reference]	Contents
Gentisic acid (C_7_H_6_O_4_)	[8]	1.5–6.7	Sinapic acid (C_11_H_12_O_5_)	[8]	0.14–2.44
Epigallocatechin (C_22_H_18_O_11_)	[8]	10.3–28.6	2,4-Dihydroxybenzoic(C_7_H_6_O_4_)	[8]	0.68–6.16
Epicatechin (C_15_H_14_O_6_)	[8]	0.08–8.4	Vanillic acid (C_8_H_8_O_4_)	[7,8]	0.10–3.91
Procyanidin B1 (C_30_H_26_O_12_)	[8]	1840–5060	Total vanillic acid	[8]	0.2–67.5
Procyanidin B2 (C_30_H_26_O_12_)	[8]	840–1460			
Procyanidin C1 (C_45_H_38_O_18_)	[8]	380–1690	Total flavonoids	[8]	6.2–30.1
Desmethylxanthohumol (C_20_H_20_O_5_)	[8]	120.0	Total flavonoids *	[8]	16.5–24.1
Isorhamnetin (C_16_H_12_O_7_)	[8]	0.5–3.3	Myricetin(C_15_H_10_O_8_)	[1,8]	0–73.3
Isoxanthohumol (C_21_H_22_O_5_)	[8]	8.0–35.2	Myricetin(C_15_H_10_O_8_) *	[8]	3.1–4.3
8-Prenylnaringenin (C_20_H_20_O_5_)	[8]	1.5–23.8	Hesperidin(C_28_H_34_O_15_)	[8]	0.5–24.9
Xanthohumol (C_21_H_22_O_5_)	[8]	85.6–480.0	Alkylresorcinols	[8]	3.2–10.3
Total trans-Stilbenes (C_14_H_12_) *	[8]	0.05–1.17	Alkylresorcinols *	[8]	2.86–3.54
trans-Resveratrol (C_20_H_20_O_9_) *	[8]	0.003–0.228	Total lignans(C_22_H_22_O_8_) *	[8]	1.25
trans-Piceid (C_20_H_22_O_8_) *	[8]	0.04–1.10			

Note: * For fresh weight the rest is dry weight.

**Table 3 molecules-29-03110-t003:** Functional and nutrient compositions of beer and barley grains.

Beer Composition			Barley Grains Composition		
(Molecular Formula)	[Reference]	Mean ± SD	(Molecular Formula)	[Reference]	Mean ± SD
Potassium (K) mg/L	[22]	543.3 ± 326.5	Potassium (K) mg/kg	[1,23]	4802 ± 1839
Sodium (Na) mg/L	[7,22]	66.4 ± 44.79	Sodium (Na) mg/kg	[1,23]	190.5 ± 104.7
Iron (Fe) mg/L	[7,22]	0.23 ± 0.19	Iron (Fe) mg/kg	[1,24]	43.4 ± 17.6
Magnesium (Mg) mg/L	[7,22]	107.0 ± 75.8	Magnesium (Mg) mg/kg	[1,23]	1250 ± 393
Calcium (Ca) mg/L	[22]	96.67 ± 15.28	Calcium (Ca) mg/kg	[1,23]	568.3 ± 235.1
Manganese (Mn) mg/L	[7,22]	0.31 ± 0.41	Manganese (Mn) mg/kg	[1,23]	29.3 ± 24.8
Zinc (Zn) mg/L	[7,22]	0.17 ± 0.12	Zinc (Zn) mg/kg	[1,24]	38.1 ± 9.3
Phosphorus (P) mg/L	[22]	361.7 ± 176.5	Phosphorus (P) mg/kg	[1,23]	2593 ± 1046
Selenium (Se) mg/L	[22]	8.33 ± 6.35	Selenium (Se) mg/kg	[24]	36.03 ± 3.62
Total prenylated flavonoids mg/L	[12]	0.0–9.5	Flavonoids mg/Kg	[20]	125.1 ± 101.4
Protein %	[22]	0.2–0.6	Protein %	[1]	14.92 ± 0.13
Polyphenols mg GAE/L	[25]	192.6 ± 8.7	Polyphenols mg GAE/L	[1,26]	2316 ± 343
FRAP mmol TE/L	[25]	1.23 ± 0.01	FRAP mmol TE/L	[1]	60.36 ± 15.48
ABTS^•+^ (C_18_H_18_N_4_O_6_S_4_) mmol TE/L	[25]	1.44 ± 0.09	ABTS (C_18_H_18_N_4_O_6_S_4_) g/L	[1]	5.87 ± 0.92
Alcohol (C_2_H_6_O) %	[22]	3.50–6.15	β-glucan (C_18_H_32_O_16_) %	[1,27]	4.61 ± 0.45
Water (H_2_O) %	[22]	88.5–97.7	Resistant starch %	[1]	3.63 ± 2.32
Total energy KJ/L	[22]	1410–2780	Arabinoxylan (C_40_H_64_O_32_) %	[1]	1.31 ± 0.73
Sugar (C_6_H_12_O_6_) %	[22]	2.1–5.3	Phenolic acids mg/100 g	[27]	414.70 ± 32.86
Iodide (I^−^) μg/L	[22]	1–8	Total flavones mg/100 g	[1,22,27]	80.64 ± 17.15
Tyrosol (C_8_H_10_O_2_) mg/L	[12]	0.2–44.4	Total alkaloid mg/100 g	[1]	25.96 ± 1.41
Hydroxy-tyrosol (C_8_H_10_O_3_) mg/L	[12]	0.0–0.1	Total anthocyanin mg/100 g	[1]	35.50 ± 23.82
Alkylresorcinols μg/L	[12]	0.02–11.0	Proanthocyanidin g/100 g	[20]	6.97 ± 3.84
Isoflavonoid (C_15_H_10_O_2_) nmol/L	[28]	0.19–14.99	Total tocols mg/100 g	[1]	5.85 ± 3.51
Daidzein (C_15_H_10_O_4_) nmol/L	[28]	0.08–2.5	Anti-oxidant activity %	[26]	41.55 ± 7.82
Genistein (C_15_H_10_O_5_) nmol/L	[28]	0.169–6.74	GABA (C_4_H_9_NO_2_) mg/100 g	[1]	8.00 ± 3.92
Biochanin A (C_16_H_12_O_5_) nmol/L	[28]	0.820–4.84	Phytosterols (C_29_H_50_O) mg/100 g	[20]	91.13 ± 21.14

**Table 4 molecules-29-03110-t004:** Functional ingredients in conventional beers.

Composition (Molecular Formula)	[Reference]	Range	Composition (Molecular Formula)	[Reference]	Range
Phenolic acid (HOC_6_H_4_SO_3_H)		mg/L Beer	Flavonoids		mg/L Beer
Gallic acid (C_7_H_6_O_5_) *	[7,8]	0.06–10.4	Kaempferol (C_15_H_10_O_6_)	[8]	0.06–16.4
Protocatechuic acid * (C_7_H_6_O_4_)	[8]	0.02–0.30	Daidzein (C_15_H_10_O_4_)	[8]	0.23–0.36
Protocatechuic acid (C_7_H_6_O_4_)	[7]	0.80–1.70	Genistein (C_15_H_10_O_5_)	[8]	0.06–0.08
p-Hydroxybenzoic acid * (C_7_H_6_O_3_)	[8]	0.38–9.04	Formononetin (C_16_H_12_O_4_)	[8]	0.17–1.30
Gentisic acid * (C_7_H_6_O_4_)	[8]	0.07–0.30	Luteolin (C_15_H_10_O_6_)	[8]	0.10–0.19
Chlorogenic acid * (C_16_H_18_O_9_)	[8]	0–2.38	Apigenin (C_15_H_10_O_5_)	[8]	0.80–0.81
2,6-dihydroxybenzoic acid * (C_7_H_6_O_4_)	[8]	2.53 ± 0.11	Myricetin (C_15_H_10_O_8_)	[8]	0.15–0.16
Vanillic * (C_8_H_8_O_4_)	[8]	0–3.6	Naringin (C_27_H_32_O_14_)	[8]	0.70–2.63
Total vanillic (C_8_H_8_O_4_)	[8]	1.17–5.45	Naringenin (C_15_H_12_O_5_)	[8]	0.06–2.34
Homo-vanillic acid * (C_9_H_10_O_4_)	[8]	0.41 ± 0.04	Phenolic alcohols		mg/L beer
Caffeic acid (C_9_H_8_O_4_)	[8]	0–2.53	Tyrosol (C_8_H_10_O_2_)	[8,12]	0.2–44.4
Total caffeic acid (C_9_H_8_O_4_)	[7,8]	0.98–6.38	Hydroxy-tyrosol (C_8_H_10_O_3_)	[8]	0.0–0.13
m-Hydroxybenzoic acid (C_7_H_6_O_3_)	[8]	0–1.03	Organic acid		mg/L beer
Syringic acid * (C_9_H_10_O_5_)	[8]	0–1.13	Lactic acid (C_3_H_6_O_3_)	[7]	28–700
Total syringic acid (C_9_H_10_O_5_)	[7,8]	0–1.23	Acetic acid (CH_3_COOH)	[7]	8–240
p-Coumaric acid * (C_9_H_8_O_3_)	[8]	0.01–5.58	Acetyl-formic acid (C_3_H_6_O)	[7]	5–330
Total p-Coumaric acid (C_9_H_8_O_3_)	[7,8]	0.30–3.10	Succinate + malic acid	[7]	61–640
Ferulic acid * (C_10_H_10_O_4_)	[7,8]	0.10–11.03	Oxalic acid (C_2_H_2_O_4_)	[7]	2–37
Total ferulic acid (C_10_H_10_O_4_)	[8]	9.97–22.60	Citric acid (C_6_H_8_O_7_)	[7]	77–590
4-hydroxyphenylacetic acid *	[8]	0.05–1.47	Sodium acetate (C_2_H_3_NaO_2_)	[21]	0.1171
Total 4-hydroxyphenylacetic acid	[8]	0.40–1.46	Na pyruvate (C_3_H_3_NaO_3_)	[21]	0.0494
Sinapic acid * (C_11_H_12_O_5_)	[8]	0.20–1.39	K D-gluconate (C_6_H_11_KO_7_)	[21]	0.0348
Total sinapic acid (C_11_H_12_O_5_)	[8]	2.19–6.16	DL-Malic acid (C_4_H_6_O_5_)	[21]	0.1867
m-Coumaric acid * (C_9_H_8_O_3_)	[8]	0.105 ± 0.006	Na citrate (Na_3_C_6_H_5_O_7_)	[21]	0.0595
Salicylic acid (C_7_H_6_O_3_)	[8]	0.19–6.66	Na DL-lactate (NaC_3_H_5_O_3_)	[21]	0.0348
o-Coumaric acid (C_9_H_8_O_3_)	[8]	0.47 ± 0.04	Prenylflavonoids		mg/L Beer
Flavonoids		mg/L Beer	8-Prenylnaringenin (C_20_H_20_O_5_)	[8]	0–0.021
Catechin (C_15_H_14_O_6_)	[7,8]	0.03–18.30	6-Geranylnaringenin	[8]	0.001–0.074
Epicatechin (C_15_H_14_O_6_)	[8]	0.02–11.50	Isoxanthohumol (C_21_H_22_O_5_)	[8]	0.04–3.44
Rutin (C_27_H_30_O_16_)	[8]	0.06–4.85	Xanthohumol (C_21_H_22_O_5_)	[8]	0.002–0.69
Quercetin (C_15_H_10_O_7_)	[8]	0.06–2.23	6-Prenylnaringenin (C_20_H_20_O_5_)	[8]	0.011–0.56
Vitamin		mg/L beer	Alkylresorcinols		mg/L Beer
Vitamin B1 (C_12_H_17_N_4_OS_+_)	[21]	0.0266	Total alkylresorcinols	[8,12]	1.01 ± 2.03
Vitamin B2 (C_17_H_20_N_4_O_6_)	[7,21]	0.26–4.03	Stilbenes (C_14_H_12_)		mg/L beer
Vitamin B3 (C_6_H_5_NO_2_)	[22]	4.30–8.15	trans-Resveratrol (C_14_H_12_O_3_)	[8]	0–0.067
Vitamin B5 (C_9_H_17_NO_5_)	[7,21]	0.033–1.065	cis-Resveratrol (C_20_H_20_O_9_)	[8]	0–0.023
Vitamin B7 (C_10_H_16_N_2_O_3S_)	[7,21]	0.008–0.022	cis-Piceid (C_20_H_22_O_8_)	[8]	0–0.024
Vitamin B6 (C_8_H_11_NO_3_)	[7,21,22]	0.05–0.505	trans-Piceid (C_20_H_22_O_8_)	[8]	0–0.009
Vitamin B8 (C_6_H_12_O_6_)	[7,21]	0–0.048	Nicotinamide (C_6_H_6_N_2_O)	[7,8]	0.01–0.30
Vitamin B9 (C_19_H_19_N_7_O_6_)	[7,21]	0–0.006	Amine		mg/L Beer
Vitamin B12 (C_63_H_88_CoN_14_O_14_P)	[22]	0.00023	Putrescine (C_4_H_12_N_2_)	[21]	0.0823
Vitamin (A, D, E, K)	[7,21]	0.398–4.984	Tyramine (C_8_H_11_NO)	[21]	1.0698
Dietary fiber		mg/L beer	Histamine (C_5_H_9_N_3_)	[21]	0.1994
Non-starchy carbohydrates	[6,29]	1442–2923	Melanoidins	[6]	0.6–103
β-glucan (C_18_H_32_O_16_)	[29]	26–99	Melanoidins/malt wort	[30]	125–225

No. * For beer, the contents of free forms of phenolic acids are reported.

## Data Availability

Data are contained within the article.

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
