# Peer review of "Physiological Mechanisms by Which the Functional Ingredients in Beer Impact Human Health"

_molecules, 2024, doi:10.3390/molecules29133110_

Round 1

Reviewer 1 Report

Comments and Suggestions for Authors

The revision is attached in a separate document.

Reviewer 1 Report

The manuscript is interesting and with the thematic scope of the Molecules journal. The manuscript which was prepared as a review is focused on the summarizing and gathering the latest knowledge about functional compounds in beer and barley which are beneficial for human health. The manuscript was prepared with using appropriate (and current) literature citations. However there are some weakness and shortcomings which should be resolved and explained during revision process. Specific comments:

Comments 1: The aim of the study is not clearly defined. Please revise.

Response 1: Thank you for pointing this out. We agree with this comment. Therefore, we have added: The aim of this review is to provide new ways for pure barley beer and new functional beer to expand the development space and its high value-added functional food, especially new insights and theoretical support for the physiological mechanisms of beer active ingredients for human health.

Comments 2: In manuscript very important issue which refers to ethanol content in beer beverages was omitted. Do you recommend to consume this alcoholic beverage by all consumers. Such information about limitations of consuming beer should be included as separated subchapter.

Response 2: Thank the reviewers for their valuable comments that played an important role in improving the quality of the paper. We have revised accordingly reviewers report: 4.2. Beer Ethanol to Human Health; Though non-alcoholic beer has antioxidation [16], improve bone health [95], maintain mood states [98], blood pressure regulation [103], promote sleep [111] and so on. The decreased risk (17%~21%) of dementia was associated with maintaining mild (<15 g per day) to moderate (15-29.9 g per day) ethanol consumption [129]; however ethanol exceeding 105g per week (i. e. <15g per day) was associated with a reduced risk of death related to heart attack, stroke, angina, or cardiovascular disease. Therefore, I recommend drinking beer with less than 30 g of ethanol per day, a 5% ethanol in beer equivalent to 600 g of beer.

Comments 3: Also in manuscript there is a lack information about sugars content in beer and problem with obesity caused by overconsumption of beer.

Response 3: Agree. We have revised accordingly reviewers report: Table 3 shows that the sugar content in beer is 2.1%~5.3%; The beer with low sugar (≤0.75 g/100 mL) and alcohol (<1.2%) contents is a functional food for diabetics [70]. Moderate intake of beer prevents lipid deposition in blood vessels and does not lead to obesity [71], however, problem with obesity caused by overconsumption of beer may be related to other high-energy foods and its improve gastrointestinal [83].

Comments 4: The picture in Figure necessarily should absolutely be removed. It is unethical to advertise beer brands in a scientific article. The same comment for Figure 2.

Response 4: Agree. We have revised accordingly reviewers report: We have deleted the advertising beer brands involved in the Figure 1 and Figure 2.

Reviewer 2 Report

Comments 1: In this narrative review, the health-promoting effects (n=28?) of a wide range of phytochemicals (isolated or synergistic) from conventional beer (CB) and its raw materials are exquisitely detailed. The manuscript is generally organized into five major sections: A) Introduction, B) Functional/nutraceutical composition of CEC and its raw materials, C) physiological mechanisms exerted and D) specific health benefits and E) structure-function claim sustentations. The narrative is supported by a very large number of references (n= 222), with small paragraphs of little depth (low inductive statements), four systematic tables and two figures. Although the information gathered in itself is unique and moderately novel (particularly because of "the new benefits", lines 206-209), some adjustments must be made to the manuscript to improve its scientific soundness and uniqueness:

Response 1: Thank the reviewers for their valuable comments that played an important role in improving the quality of the paper. (1) “3. Functional Ingredients in Beer” was merged into “2. Functional Ingredients of Beer and Its Raw Materials”, the serial number is adjusted accordingly. (2) “4. The Health Effects of Raw Materials” becomes “3. The Health Effects of Raw Materials”, the serial number is adjusted accordingly. (3) “5 Action of Beer on Human Health” becomes “4. Structure-function Claim Sustentation of Beer”, the serial number is adjusted accordingly. (4) “6. Molecular Mechanism of Beer to Human Health” becomes “5. Physiological Mechanism of Beer Affecting Human Health”, the serial number is adjusted accordingly. “7. Conclusion and Future Perspectives” becomes “6. Conclusion and Future Perspectives”.

Comments 2: General. A) Reading and understanding of the manuscript will improve if the next version (v2) is reviewed by a native English speaker or by a formal translation agency. B) Instead of leaving the meaning of each abbreviation at the end of the manuscript (Line 897-898), it is requested to leave the meaning within the text the first time it is mentioned, as required in authors ‘guidelines.

Response 2: Thank the reviewers for their valuable comments that played an important role in improving the quality of the paper. We have revised accordingly reviewers report: Reading and understanding of the manuscript will improve by a formal translation agency. It has been revised according to the review opinions that each abbreviation at the end of the manuscript (Line 897-898), it is requested to leave the meaning within the text first time it is mentioned, as required in authors ‘guidelines.

Comments 3: Title. A) “Actional mechanism” should be changed for a more proper term such as “physiological mechanism” or “health-promoting effects” B) Much more emphasis should be placed on enhancing the health-promoting benefits of conventional beer vs. the artisanal and/or the "functional" ones, this must be perfectly clear, in light of the fact that all of them vary not only the profile of functional phytochemicals but also the alcohol content and other psychotropic bioactives (https://doi.org/10.3390/beverages6030051). Barley should be minimized form here and throughout de manuscript if it is not the main theme. 

Response 3: Agree. We have revised accordingly reviewers report: Title“Actional Mechanism of Functional Ingredients in Beer and Barley for Human Health”becomes“Physiological Mechanisms by which the Functional Ingredients in Beer Impact Human Health”We have enhanced the health-promoting benefits of conventional beer vs functional beer, such as “A general wish for healthier lifestyles has resulted in increased demand for functional beers with health benefits and sensory adjustments of classical beer, which has broadened the market for the brewing industry [5]…The potential for functional beer expansions is endless when it comes to combining beer with herbs, spices, and other functional compounds [5]… (http://dx.doi.org/10.3390/beverages6030051).

Comments 4:  Abstract. It seems not to be focused on the structure and type of information presented. It is advisable to reconstruct accordingly.

Response 4: Agree. The abstract has been rewritten, see the paper for details.

Comments 5:   Introduction. OK. Sections. Reading all the sections is difficult not only because of the grammar and syntax but also because the paragraphs are not constructed in an "effective" way (check and reconstruct: http://purdueglobalwriting.center/how-to-write-an-effective-paragraph/ ).

Response 5: Agree. Thank the reviewers for their valuable comments that played an important role in improving the quality of the paper:We have checked and reconstructed / modified over 50 paragraphs in an "effective" way (See Molecules-3027406-2),according to reviewer: http://purdueglobalwriting.center/how-to-write-an-effective-paragraph/

Comments 6:  Figures & Tables. A) Figures should be provided with enough resolution (≥300 dpi) and tables should be formatted according to this journal´s guidelines. B) Table 1-3: i) next to the name of the phytochemical and include it below the table. Each table and figure must be understood without consulting the text, ii) Arrange it in three columns (phytochemical, hops & barley) to visualize whether the phytochemical comes from one (e.g. gentisic or sinapic acids) or both sources (e.g. caffeic acid). C) Table 4. Eliminate intermediate lines according to authors ‘guidelines.

Response 6: Agree. We have revised accordingly reviewers report: A) Figures have enough resolution (≥300 dpi) and tables should be formatted according to this journal´s guidelines. B)Table 1-3: i) We supplement the molecular formula of the phytochemical name whenever possible in the table. ii) We divided Table 1 into two tables: Table 1. Fourteen phenolic compounds in common beer from hops and barley [English name/molecular formula/Phytochemical name, hops, barley, Reference]; Table 2. Phenolic compounds in beer from hops or barley. C)Table 4. We Eliminate intermediate lines according to authors ‘guidelines. In addition, Tables 2,3 and 4 are changed to tables 3,4 and 5, respectively

Comments 7: References. A) Reduce the number of references to say 150 max. B) reduce self-citing (references: 1, 19, 26, 27, 36, 48, 51, 52, 55, 56, 64, 86, 151, 176, 200, 219) up to 4 C) Please check once again the references´ format according to this journal´s guidelines.

Response 7: Agree. We have revised accordingly reviewers report: We removed 27 reference contributions, especially self-citing (57131921262734363948515255565764717686137149162176177178191200).

4. Response to Comments on the Quality of English Language

Point 1: The reviewers indicate that your manuscript may require moderate or extensive English revisions.

Response 1: This review paper has undergone English language editing by MDPI

5. Additional clarifications

; The text has been checked for correct use of grammar and common technical terms, and edited to a level suitable for reporting research in a scholarly journal(see English-Editing-Certificate-81797).

Reviewer 2 Report

Comments and Suggestions for Authors

In this narrative review, the health-promoting effects (n=28?) of a wide range of phytochemicals (isolated or synergistic) from conventional beer (CB) and its raw materials are exquisitely detailed. The manuscript is generally organized into five major sections: A) Introduction, B) Functional/nutraceutical composition of CEC and its raw materials, C) physiological mechanisms exerted and D) specific health benefits and E) structure-function claim sustentations. The narrative is supported by a very large number of references (n= 222), with small paragraphs of little depth (low inductive statements), four systematic tables and two figures. Although the information gathered in itself is unique and moderately novel (particularly because of "the new benefits", lines 206-209), some adjustments must be made to the manuscript to improve its scientific soundness and uniqueness:

·         General. A) Reading and understanding of the manuscript will improve if the next version (v2) is reviewed by a native English speaker or by a formal translation agency. B) Instead of leaving the meaning of each abbreviation at the end of the manuscript (Line 897-898), it is requested to leave the meaning within the text the first time it is mentioned, as required in authors ‘guidelines.

·         Title. A) “Actional mechanism” should be changed for a more proper term such as “physiological mechanism” or “health-promoting effects” B) Much more emphasis should be placed on enhancing the health-promoting benefits of conventional beer vs. the artisanal and/or the "functional" ones, this must be perfectly clear, in light of the fact that all of them vary not only the profile of functional phytochemicals but also the alcohol content and other psychotropic bioactives (doi: http://dx.doi.org/10.3390/beverages6030051 , https://doi.org/10.3390/beverages6030051). Barley should be minimized form here and throughout de manuscript if it is not the main theme.

·         Abstract. It seems not to be focused on the structure and type of information presented. It is advisable to reconstruct accordingly.

·         Introduction. OK.

·         Sections. Reading all the sections is difficult not only because of the grammar and syntax but also because the paragraphs are not constructed in an "effective" way (check and reconstruct: http://purdueglobalwriting.center/how-to-write-an-effective -paragraph/ ).

·         Figures & Tables. A) Figures should be provided with enough resolution (≥300 dpi) and tables should be formatted according to this journal´s guidelines. B) Table 1-3: i) next to the name of the phytochemical and include it below the table. Each table and figure must be understood without consulting the text, ii) Arrange it in three columns (phytochemical, hops & barley) to visualize whether the phytochemical comes from one (e.g. gentisic or sinapic acids) or both sources (e.g. caffeic acid). C) Table 4. Eliminate intermediate lines according to authors ‘guidelines.

·         References. A) Reduce the number of references to say 150 max. B) reduce self-citing (references: 1, 19, 26, 27, 36, 48, 51, 52, 55, 56, 64, 86, 151, 176, 200, 219) up to 4 C) Please check once again the references´ format according to this journal´s guidelines.

Comments on the Quality of English Language

Moderate editing is needed

Reviewer 2 Report

Comments 1: In this narrative review, the health-promoting effects (n=28?) of a wide range of phytochemicals (isolated or synergistic) from conventional beer (CB) and its raw materials are exquisitely detailed. The manuscript is generally organized into five major sections: A) Introduction, B) Functional/nutraceutical composition of CEC and its raw materials, C) physiological mechanisms exerted and D) specific health benefits and E) structure-function claim sustentations. The narrative is supported by a very large number of references (n= 222), with small paragraphs of little depth (low inductive statements), four systematic tables and two figures. Although the information gathered in itself is unique and moderately novel (particularly because of "the new benefits", lines 206-209), some adjustments must be made to the manuscript to improve its scientific soundness and uniqueness:

Response 1: Thank the reviewers for their valuable comments that played an important role in improving the quality of the paper. (1) “3. Functional Ingredients in Beer” was merged into “2. Functional Ingredients of Beer and Its Raw Materials”, the serial number is adjusted accordingly. (2) “4. The Health Effects of Raw Materials” becomes “3. The Health Effects of Raw Materials”, the serial number is adjusted accordingly. (3) “5 Action of Beer on Human Health” becomes “4. Structure-function Claim Sustentation of Beer”, the serial number is adjusted accordingly. (4) “6. Molecular Mechanism of Beer to Human Health” becomes “5. Physiological Mechanism of Beer Affecting Human Health”, the serial number is adjusted accordingly. “7. Conclusion and Future Perspectives” becomes “6. Conclusion and Future Perspectives”.

Comments 2: General. A) Reading and understanding of the manuscript will improve if the next version (v2) is reviewed by a native English speaker or by a formal translation agency. B) Instead of leaving the meaning of each abbreviation at the end of the manuscript (Line 897-898), it is requested to leave the meaning within the text the first time it is mentioned, as required in authors ‘guidelines.

Response 2: Thank the reviewers for their valuable comments that played an important role in improving the quality of the paper. We have revised accordingly reviewers report: Reading and understanding of the manuscript will improve by a formal translation agency. It has been revised according to the review opinions that each abbreviation at the end of the manuscript (Line 897-898), it is requested to leave the meaning within the text first time it is mentioned, as required in authors ‘guidelines.

Comments 3: Title. A) “Actional mechanism” should be changed for a more proper term such as “physiological mechanism” or “health-promoting effects” B) Much more emphasis should be placed on enhancing the health-promoting benefits of conventional beer vs. the artisanal and/or the "functional" ones, this must be perfectly clear, in light of the fact that all of them vary not only the profile of functional phytochemicals but also the alcohol content and other psychotropic bioactives (https://doi.org/10.3390/beverages6030051). Barley should be minimized form here and throughout de manuscript if it is not the main theme. 

Response 3: Agree. We have revised accordingly reviewers report: Title“Actional Mechanism of Functional Ingredients in Beer and Barley for Human Health”becomes“Physiological Mechanisms by which the Functional Ingredients in Beer Impact Human Health”We have enhanced the health-promoting benefits of conventional beer vs functional beer, such as “A general wish for healthier lifestyles has resulted in increased demand for functional beers with health benefits and sensory adjustments of classical beer, which has broadened the market for the brewing industry [5]…The potential for functional beer expansions is endless when it comes to combining beer with herbs, spices, and other functional compounds [5]… (http://dx.doi.org/10.3390/beverages6030051).

Comments 4:  Abstract. It seems not to be focused on the structure and type of information presented. It is advisable to reconstruct accordingly.

Response 4: Agree. The abstract has been rewritten, see the paper for details.

Comments 5:   Introduction. OK. Sections. Reading all the sections is difficult not only because of the grammar and syntax but also because the paragraphs are not constructed in an "effective" way (check and reconstruct: http://purdueglobalwriting.center/how-to-write-an-effective-paragraph/ ).

Response 5: Agree. Thank the reviewers for their valuable comments that played an important role in improving the quality of the paper:We have checked and reconstructed / modified over 50 paragraphs in an "effective" way (See Molecules-3027406-2),according to reviewer: http://purdueglobalwriting.center/how-to-write-an-effective-paragraph/

Comments 6:  Figures & Tables. A) Figures should be provided with enough resolution (≥300 dpi) and tables should be formatted according to this journal´s guidelines. B) Table 1-3: i) next to the name of the phytochemical and include it below the table. Each table and figure must be understood without consulting the text, ii) Arrange it in three columns (phytochemical, hops & barley) to visualize whether the phytochemical comes from one (e.g. gentisic or sinapic acids) or both sources (e.g. caffeic acid). C) Table 4. Eliminate intermediate lines according to authors ‘guidelines.

Response 6: Agree. We have revised accordingly reviewers report: A) Figures have enough resolution (≥300 dpi) and tables should be formatted according to this journal´s guidelines. B)Table 1-3: i) We supplement the molecular formula of the phytochemical name whenever possible in the table. ii) We divided Table 1 into two tables: Table 1. Fourteen phenolic compounds in common beer from hops and barley [English name/molecular formula/Phytochemical name, hops, barley, Reference]; Table 2. Phenolic compounds in beer from hops or barley. C)Table 4. We Eliminate intermediate lines according to authors ‘guidelines. In addition, Tables 2,3 and 4 are changed to tables 3,4 and 5, respectively

Comments 7: References. A) Reduce the number of references to say 150 max. B) reduce self-citing (references: 1, 19, 26, 27, 36, 48, 51, 52, 55, 56, 64, 86, 151, 176, 200, 219) up to 4 C) Please check once again the references´ format according to this journal´s guidelines.

Response 7: Agree. We have revised accordingly reviewers report: We removed 27 reference contributions, especially self-citing (57131921262734363948515255565764717686137149162176177178191200).

4. Response to Comments on the Quality of English Language

Point 1: The reviewers indicate that your manuscript may require moderate or extensive English revisions.

Response 1: This review paper has undergone English language editing by MDPI

5. Additional clarifications

; The text has been checked for correct use of grammar and common technical terms, and edited to a level suitable for reporting research in a scholarly journal(see English-Editing-Certificate-81797).